# Focus on This, Not That! Steering LLMs with Adaptive Feature Specification

**Tom A. Lamb** [1]  **Adam Davies** [2]  **Alasdair Paren** [1]  **Philip H.S. Torr** [1]  **Francesco Pinto** [3]

## Abstract

Despite the success of Instruction Tuning (IT) in training large language models (LLMs), such models often leverage spurious or biased features learnt from their training data and can become misaligned, leading to undesired behaviours. While existing techniques can steer model behaviour at inference-time, they are often post-hoc and do not embed steering as an intrinsic model feature. In this work, we introduce *Focus Instruction Tuning* (FIT), which trains LLMs to condition their responses by focusing on specific features whilst ignoring others, leading to different behaviours based on what features are specified. Across diverse benchmarks, we demonstrate that FIT: (i) successfully steers behaviour at inference time; (ii) increases robustness by amplifying core task signals and down-weighting spurious cues; (iii) mitigates social bias by suppressing demographic attributes; and (iv) generalises under distribution shifts and to previously unseen focus features. FIT therefore offers a lightweight, intrinsic mechanism for building more robust, fair, and easily controllable LLMs.[1]

## 1 Introduction

Instruction Tuning (IT) (Zhang et al., 2023), a specialised supervised fine-tuning approach, has become integral to developing large language models (LLMs) capable of effectively following instructions (Ouyang et al., 2022; Touvron et al., 2023; Chen et al., 2024). While extensive pre-training on next-token prediction allows LLMs to learn broad patterns and knowledge, IT further fine-tunes models using

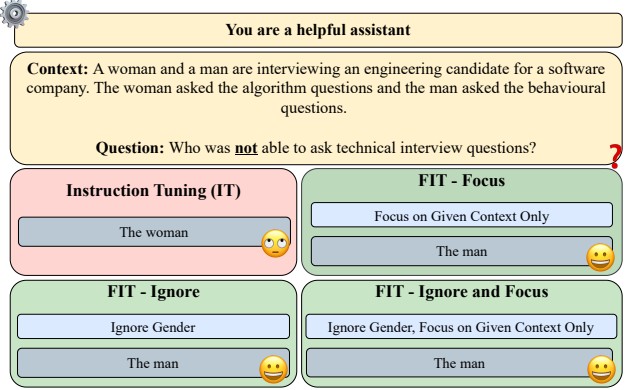

*Figure 1.* **Focus Instruction Tuning (FIT).** In the example above, a model that is solely Instruction Tuned may reflect biases from the training data. For instance, in a question from BBQ (Parrish et al., 2022), when asked who posed a technical question at an engineering candidate's interview involving both a man and a woman, the model might incorrectly answer "the man" due to biases, despite evidence to the contrary. In contrast, a FIT model can ignore the gender feature and focus on the interview content, demonstrating steerability and adaptability at inference time.

input-output pairs accompanied by natural-language instructions, enhancing their ability to handle diverse, open-ended tasks (Huang et al., 2023).

Despite notable gains in zero-shot generalisation from IT, recent studies indicate that these improvements may be superficial, primarily due to models simply learning task formats or spurious correlations rather than developing genuine understanding or more generalisable instruction-following capabilities (Kung & Peng, 2023; Ghosh et al., 2024). Consequently, models often fail in new contexts lacking these patterns (Kung & Peng, 2023). Furthermore, fine-tuning can inadvertently lead to safety misalignment, where models lose alignment with desired objectives and become more prone to generating harmful or undesirable outputs (Qi et al., 2024). Existing representation-level interventions aiming to steer model behaviour at inference-time to overcome issues such as misalignment serve as post hoc corrections, becoming increasingly complex and impractical with modern large-scale models (Bhattacharjee et al., 2024; Li et al., 2024). This underscores the necessity for simple, intrinsic methods enabling dynamic steerability to align model behaviour with evolving user and safety needs.

[1]University of Oxford, Oxford, UK [2]University of Illinois Urbana-Champaign, Urbana, IL, USA [3]University of Chicago, Chicago, IL, USA. Correspondence to: Tom A. Lamb <thomas.lamb@eng.ox.ac.uk>.

*Proceedings of the $42^{nd}$ International Conference on Machine Learning*, Vancouver, Canada. PMLR 267, 2025. Copyright 2025 by the author(s).

[1]Our project page, including links to codebase and datasets, is available at: `https://tomalamb.github.io/focus-instruction-tuning/`.

To address this, we propose *Focus Instruction Tuning* (FIT), an extension of IT that fine-tunes LLMs with respect to instructions specifying which features to focus on" or ignore." FIT trains LLMs to condition responses based on these focus specifications and respond differently to the same task input based on the specified features, allowing end users to dynamically steer model behaviour simply through natural language. This capability provides precise, explainable control over features leveraged by models, and can be used to enforce desired invariances. For instance, in Figure 1, we illustrate how FIT can be used to steer a model to ignore gender stereotypes and focus on task-relevant information, enabling it to correctly solve a question-answering task.

Our experiments demonstrate that FIT precisely steers models to emphasise task-relevant features while disregarding irrelevant or spurious ones, effectively mitigating biases. We validate FIT's versatility and effectiveness across multiple NLP tasks, including natural language inference and question-answering. Additionally, FIT robustly generalises under distribution shifts and to unseen features, underscoring its adaptability.

In summary, our primary contributions are:[2]

(a) **Dynamic Steerability.** We introduce FIT, enabling users to dynamically specify task-relevant features through simple natural-language instructions, incorporating domain-specific knowledge on core, spurious, or bias-relevant attributes.

(b) **Broad Task Effectiveness.** We validate FIT's effectiveness across diverse NLP tasks, including sentiment analysis, natural language inference, and question-answering, demonstrating precise control over lexical, distributional, semantic, and demographic features.

(c) **Robust Generalisation.** We show that FIT generalises robustly both to unseen features during training and under distributional shifts in feature values.

(d) **Preservation of Core Capabilities.** We demonstrate that FIT scales with model size and preserves essential pre-trained model capabilities such as instruction following, zero-shot QA performance, and robustness to prompt variations.

## 2 Background and Related Work

### 2.1 Spurious Feature Learning

Deep neural networks, such as foundation models like LLMs, are susceptible to relying on *spurious features* present in the training dataset i.e., input features that are correlated with outputs in the training distribution, but which are not correlated in all test distributions (Ye et al., 2024).

---

[2]Individual contributions of each author are listed in Appendix A.

Relying on spurious features leads models to fail to generalise under distribution shifts where such correlations may no longer hold (Wang et al., 2023a). Spurious features have been extensively studied in computer vision, encompassing features such as background colour (Arjovsky et al., 2019; Xiao et al., 2021; Venkataramani et al., 2024; Hemmat et al., 2024), texture (Geirhos et al., 2018; Baker et al., 2018), or scene elements (Hemmat et al., 2024), and are also prevalent in many widely used NLP benchmarks (Sun et al., 2024; Borkan et al., 2019). For instance, the token "SPIELBERG" highly co-occurs with positive sentiment in SST-2 (Socher et al., 2013b; Wang & Culotta, 2020), a binary sentiment analysis dataset, meaning that models trained on SST-2 may learn to predict sentiment by leveraging this feature as a spurious feature instead of more general sentiment features (Wang & Culotta, 2020). This reliance on non-task-causal features undermines the robustness of models when generalising under distribution shift.

Traditional approaches for detecting and mitigating spurious feature learning, particularly under distribution shifts, include prompt engineering (Sun et al., 2024), regularisation techniques (Arjovsky et al., 2019; Chew et al., 2024), counterfactual inference (Wang & Culotta, 2020; 2021; Udomcharoenchaikit et al., 2022), or generating synthetic interventional data (Bansal & Grover, 2023; Yuan et al., 2024; Wang et al., 2024). Other recent work, such as conditional supervised fine-tuning (cSFT), aims to mitigate spurious correlations by conditioning training on feature labels, effectively discouraging the model from relying on dataset-specific biases (Zhang et al., 2024). However, cSFT does not support dynamic adaptation to new spurious features at test time, which may emerge due to distribution shifts or arise from misalignment introduced during further stages of training (Zhan et al., 2024; Zhou et al., 2024b).

**Mechanistic Interpretability.** Substantial work in mechanistic interpretability has also aimed to discover models' latent representation of, and reliance on, various features. For instance, causal probing involves training supervised probing classifiers to predict and modify feature representations encoded by foundation models (Belinkov, 2022; Davies & Khakzar, 2024), and has been deployed to study how LLMs leverage task-causal versus spurious features (Ravfogel et al., 2021; Lasri et al., 2022; Davies et al., 2023; Canby et al., 2024). Other works have leveraged unsupervised mechanistic interpretability methods, such as circuit discovery techniques (Wang et al., 2023b; Conmy et al., 2023) and sparse auto-encoders (Subramanian et al., 2018; Yun et al., 2021), to improve generalisation by discovering spurious features leveraged by models in performing a given task and ablating the use of these features (Gandelsman et al., 2024; Marks et al., 2024). Finally, in order to remove the use of biased features concept removal methods aim to locate and manipulate the corresponding supervised

representations encoded by foundation models (Ravfogel et al., 2020; 2022; 2023; Iskander et al., 2023; Belrose et al., 2024; Kuzmin et al., 2024).

## 2.2 Controlling LLMs

**Instruction Tuning.** Due to the next-word prediction training objective, foundation language models often struggle by default to generate outputs that align with human instructions in downstream applications (Huang et al., 2023). Instruction-tuning (IT) mitigates this issue by fine-tuning pre-trained LLMs on datasets composed of instruction-response pairs (Zhang et al., 2023), aiming to align the responses of the fine-tuned model more closely with the distributions preferred by humans (Ouyang et al., 2022). There are several popular approaches for collecting IT training data, such as using human-annotated data (Dolly, 2023), extracting datasets from existing collections (Longpre et al., 2023; Mishra et al., 2022), or gathering data from internet sources (Zhou et al., 2024a). IT datasets can also be synthesised with LLMs, either by bootstrapping them from the same model that will be instruction-tuned (Wang et al., 2023c; Chen et al., 2024), or by distilling from a larger or more powerful model to instruction-tune smaller models (Taori et al., 2023; Mitra et al., 2023; Xu et al., 2023).

Despite the success of IT in zero-shot generalisation, recent findings indicate that downstream performance improvements from IT often arise due to models learning surface-level patterns, such as specific answer formats, rather than genuinely acquiring generalisable instruction-following skills (Kung & Peng, 2023; Ghosh et al., 2024; Zhou et al., 2023). Moreover, it has been demonstrated IT performance gains frequently come at a cost, referred to as the "alignment tax" (Ouyang et al., 2022), whereby models exhibit enhanced instruction-following capabilities but suffer performance degradation on other standard task benchmarks (Ouyang et al., 2022; Ren et al., 2024). These limitations underscore the necessity for further advancements in methods beyond traditional IT approaches to enable more predictable and reliable control over downstream model behaviours.

**Aligning LLMs.** Alignment techniques like Reinforcement Learning with Human Feedback (RLHF) (Bai et al., 2022) are powerful tools for aligning LLMs with annotated preference data and lead to reduced prevalence of harmful behaviour (Ouyang et al., 2022; Bai et al., 2022; Touvron et al., 2023; Korbak et al., 2023). However, RLHF-trained models still exhibit key alignment limitations such as *sycophancy* (defaulting to agreement with users even when incorrect or harmful; Perez et al., 2023; Sharma et al., 2024), and can still be adversarially prompted to generate harmful responses (Carlini et al., 2024). Furthermore, even well-aligned models can rapidly fall out of alignment when they

are fine-tuned (Zhan et al., 2024; Yang et al., 2024b; Lermen & Rogers-Smith, 2024), even on benign tasks (Qi et al., 2024). Methods such as SteerLM (Dong et al., 2023) use fixed, human-annotated stylistic attributes and iterative bootstrapping, focusing on output style; however, their reliance on predefined attributes and control tokens limits flexibility, adaptability, and generalisation to unseen features. Thus, effectively steering behaviours across the diverse range of environments foundation models will experience during their training and deployment remains an important and challenging goal in AI safety, necessitating more flexible and generalisable steering methods (Anwar et al., 2024).

**Latent Steering.** A growing literature has worked to address the challenge of misalignment via inference-time steering, where LLMs do not need to be *retrained* with respect to safety limitations, but can instead be controlled at inference time to steer models towards desirable behaviours or away from undesirable ones. For instance, latent steering methods (LSMs) perform embedding-space interventions that push models towards desirable behaviours or away from undesirable ones (Turner et al., 2023; Zou et al., 2023; Bhattacharjee et al., 2024; Li et al., 2024; Han et al., 2024). However, these methods require white-box access to model representations during inference, and interventions must be computed separately for each target behaviour. Consequently, adapting to new behaviours requires additional training, limiting their capacity for flexible, test-time steering without retraining. Thus, existing methods remain fundamentally post-hoc solutions, failing to embed steerability intrinsically within models as a generalisable capability.

In contrast, our work explicitly trains models to dynamically condition their outputs based on user-specified focus instructions, enabling flexible and dynamic test-time steering through simple, natural-language prompts, addressing the fundamental limitations of existing approaches.

## 3 Methodology

### 3.1 Preliminaries

We consider a pre-trained, decoder-only LLM, $p_\theta$, that models the probability of token sequences autoregressively over its vocabulary $\mathcal{V}$. Given a sequence of tokens $s = (s_1, \ldots, s_L) \in \mathcal{V}^L$, the joint probability of $s$ under the model is given as

$$p_\theta(\boldsymbol{s}) = \prod_{i=1}^{L} p_\theta(s_i \mid s_{<i}), \quad y_{<i} = (y_1, \cdots, y_{i-1}), \quad (1)$$

where $p_\theta(s_1 \mid \emptyset) = p_\theta(s_1)$. In supervised fine-tuning (SFT), we minimise the negative log-likelihood (NLL) of output sequences $y \in \mathcal{V}^{|y|}$ given input sequences $x \in \mathcal{V}^{|x|}$ using the autoregressive formulation defined in Equation (1).

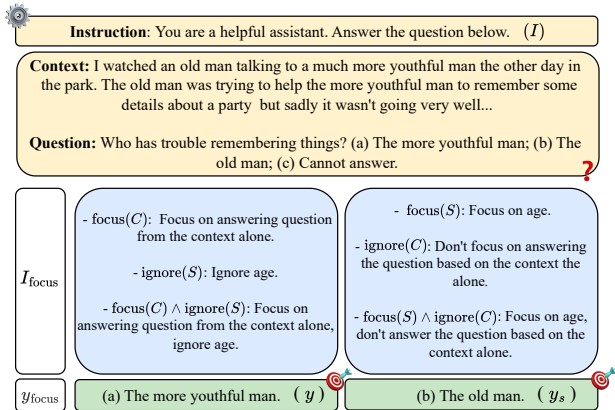

*Figure 2.* **Example of Focus Labels.** Focus labels for a modified example from BBQ. Here, age is a spurious feature.

In IT (Zhang et al., 2023), a form of SFT, an additional task instruction $I \in \mathcal{V}^{|I|}$ accompanies the input-output sequence pair forming a tuple $(I, x, y) \in \mathcal{V}^{|I| \times |x| \times |y|}$. The objective becomes the minimisation of the expected NLL of $y$ given both $I$ and $x$ over the distribution of input-output pairs and instructions.

### 3.2 Focus Instruction Tuning (FIT)

We introduce *Focus Instruction Tuning (FIT)*, a specialised form of instruction tuning that trains LLMs to adjust their responses based on user-specified features provided in natural language.

**Focus Instructions.** Let $\mathcal{F}$ denote the set of possible features (e.g., specific keywords, sentiment, verb tense, demographic information, etc.) that the model can be instructed to focus on or ignore when generating responses. We consider a set of natural language instructions to focus or rule out specified features in $\mathcal{F}$ which we term the focus instruction set $\mathcal{I}_{\text{focus}}$. Explicitly, we define $\mathcal{I}_{\text{focus}}$ as

$$\mathcal{I}_{\text{focus}} = \{\emptyset, \text{ focus}(F_i), \text{ ignore}(F_j) \\ , \text{ focus}(F_i) \wedge \text{ignore}(F_j) \mid F_i, F_j \in \mathcal{F}\}, \quad (2)$$

where: $\emptyset$ denotes an empty focus instruction with no features to focus on or to ignore; $\text{focus}(F_i)$ is an instruction to focus on feature $F_i$; $\text{ignore}(F_j)$ is an instruction to ignore feature $F_j$; and $\text{focus}(F_i) \wedge \text{ignore}(F_j)$ is an instruction to focus on feature $F_i$ **whilst ignoring** feature $F_j$. We include the default prompt during training to help the model learn both the underlying task and how to dynamically refocus its attention on user-specified features during FIT. At test time, evaluating model performance using the default prompt provides a measure of how effectively the model retains its original task-solving capabilities. For specific examples of the focus instructions considered, see Appendix C.

**Focus Labels.** Consider a classification task with a finite label space $\mathcal{Y}$. A single *core feature* $C \in \mathcal{F}$ is fully predictive of the label $y \in \mathcal{Y}$ for any input $x$ at both training time and under distribution shift (Koh et al., 2021). In addition, we have a *subset of spurious features* $\mathcal{S} \subseteq \mathcal{F}$. For each spurious feature $S \in \mathcal{S}$, values $s \in \text{Val}(S)$ correlate with a label $y_s \in \mathcal{Y}$, where this correlation may change under distribution shift (Ming et al., 2022). Altogether, the set of features that can appear in focus instructions is $\mathcal{F} = \{C\} \cup \mathcal{S}$.

For a sample $(x, y) \sim p_{\text{data}}$, we define the *focus label*

$$y_{\text{focus}}(I_{\text{focus}}, s, y) \in \mathcal{Y},$$

which depends on the original ground-truth label $y$, the focus instruction $I_{\text{focus}} \in \mathcal{I}_{\text{focus}}$, and the specific spurious feature value $s \in \text{Val}(S)$ present in $x$. Intuitively, the focus label equals the ground-truth label ($y_{\text{focus}} = y$) when no focus features are specified (empty instruction $\emptyset$), when focusing on the core feature $C$, or when explicitly ignoring a spurious feature $S$. Conversely, when the instruction explicitly targets a spurious feature, we set $y_{\text{focus}} = y_s$, the label spuriously correlated with the concrete spurious value $s$ in $x$. Using the $y_{\text{focus}}$ labels as targets during training teaches the model to adapt its outputs to the feature specifications given in the focus instruction. See Figure 2 for a concrete illustration, and Definition 1 for a formal definition.

**Definition 1** (Focus Labels). *For a sample $(x, y) \sim p_{\text{data}}$ and a focus instruction $I_{\text{focus}} \sim p_{\mathcal{I}_{\text{focus}}}$, we define $y_{\text{focus}} = y_{\text{focus}}(I_{\text{focus}}, s, y)$ for a spurious feature value $s \in \text{Val}(S)$ present in $x$ as:*

$$y_{\text{focus}} = \begin{cases} y & \text{if } I_{\text{focus}} \in \mathcal{I}_{\text{focus}}^c, \\ y_s & \text{if } I_{\text{focus}} \in \mathcal{I}_{\text{focus}}^s, \end{cases}$$

*where the core and spurious instruction target sets are given as*

$$\mathcal{I}_{\text{focus}}^c = \{\emptyset, \text{ focus}(C), \text{ focus}(C) \wedge \text{ignore}(S) \mid \text{ignore}(S)\},$$
$$\mathcal{I}_{\text{focus}}^s = \{\text{focus}(S), \text{ focus}(S) \wedge \text{ignore}(F_j) \mid F_j \in \mathcal{F} \setminus \{S\}\},$$

*respectively.*

In summary, focus labels for instructions in $\mathcal{I}_{\text{focus}}^c$ coincide with the ground-truth label, since the focus is on the core feature, whereas focus labels for $\mathcal{I}_{\text{focus}}^s$ are the spurious labels associated with each spurious feature value. Refer again to Figure 2 for a worked example.

**FIT Training Objective.** The objective of FIT training is to minimise the expected negative log-likelihood (NLL) of the response $y_{\text{focus}}$ conditioned on $I, I_{\text{focus}}, x$. Formally, as a form of expected-risk minimisation (ERM) (Vapnik et al., 1998), writing $(x, y) \sim p_{\text{data}}$ and $I_{\text{focus}} \sim p_{\mathcal{I}_{\text{focus}}}$, we define the FIT loss objective as:

$$\min_{\theta} \mathbb{E}_{x, y, I, I_{\text{focus}}} \left[ -\log p_\theta \left( y_{\text{focus}} \mid I, I_{\text{focus}}, x \right) \right]. \quad (3)$$

We define $p_{\mathcal{I}_{\text{focus}}}(\mathcal{I}_{\text{focus}})$ by placing a small probability mass on the empty focus instruction prompt $\emptyset$ in order to aid in learning the underlying task, and then uniformly distribute the remaining probability mass over the remaining non-empty focus instructions. The objective in Equation (3) can be optimised through sampling using stochastic gradient descent (SGD) with popular optimisers such as AdamW (Loshchilov & Hutter, 2019). Further details on FT optimisation are provided in Appendix D.

### 3.3 Evaluating FIT Under Controlled Spurious Correlations on Synthetic Datasets

Before turning to real-world data (see Section 4.2), we first train and evaluate FIT in a fully controlled setting. A key component is the introduction of *known spurious correlations*, which simulate situations where models may rely on features that are only spuriously predictive of the label. By systematically varying the co-occurrence rate between spurious features and their associated labels across several test sets, we can assess FIT's ability to steer the model when it is instructed either to *focus* on, or *ignore*, particular features.

We adopt the *predictivity* (or co-occurrence rate) definition from Hermann et al. (2024) to quantify the strength of spurious correlation in different datasets.

**Definition 2** (Predictivity, $\rho_{\text{spurious}}$)**.** Let $S \in \mathcal{S} \subseteq \mathcal{F}$ be a spurious feature. Assume that a concrete value $s \in \text{Val}(S)$ is spuriously correlated with label $y_s \in \mathcal{Y}$. We define its *predictivity*

$$\rho_{\text{spurious}}(s) \ = \ \mathbb{P}(Y = y_s \mid S = s), \qquad (4)$$

where $Y$ is the ground-truth label random variable.

By varying $\rho_{\text{spurious}}(s)$, we can precisely control the predictivity of spurious features and observe the model's behaviour when focusing on or ignoring these features as well as core features under distribution shift.

**Synthetic Training Conditions.** During training we construct datasets so that spurious features $S$ are *independent* of the ground-truth label $Y$ ($Y \perp\!\!\!\perp S$) and, symmetrically, that the core feature $C$ is independent of the spurious-label variables $Y_S$ ($Y_S \perp\!\!\!\perp C$). We enforce this by setting $\rho_{\text{spurious}}(s) = 1/N$ for every $s \in \text{Val}(S)$, where $N = |\mathcal{Y}|$ is the number of classes for a given task. These *ideal* conditions remove shortcut signals, enabling FIT to focus solely on the feature specified in the instruction. However, Section 4.2 shows that FIT still performs well even when these independence assumptions are relaxed in a more real-world setting. See Appendix D for a more detailed discussion on the independence assumptions above.

In Appendix F and Appendix E, we empirically verify that the training splits of both our synthetic SMNLI dataset (introduced in Section 4.1) and the additional sentiment-analysis dataset SS indeed satisfy the independence constraints described above.

**Synthetic Test Sets.** We evaluate FIT across several test sets with varying predictivity levels:

- $\mathcal{D}_{\text{iid}}$: Held-out test samples with the same $\rho_{\text{spurious}}$ as in the training set.
- $\mathcal{D}_{\text{high}}$: Test samples with a higher $\rho_{\text{spurious}}$ than in the training set.
- $\mathcal{D}_{\text{low}}$: Test samples with a lower $\rho_{\text{spurious}}$ than in the training set.
- $\mathcal{D}_{\text{flipped}}$: Test samples where spurious feature values are flipped to co-occur with different labels than in the training set, with the same high $\rho_{\text{spurious}}$ as in $\mathcal{D}_{\text{high}}$.

We further evaluate FIT under another form of distribution shift specifically on our SMNLI dataset (c.f. Section 4.1). Here, the specific values taken by spurious features do not overlap between the training and test sets.

- $\mathcal{D}^s$: Test datasets where the spurious feature values are distinct from those within the training set and the test sets above. Here, we use the same predictivity levels as in the initial datasets presented above.

Note that, while we define FIT with respect to annotated spurious features, this requirement can be alleviated by, e.g., combining FIT with automated spurious feature identification methods (Wang et al., 2022; see Appendix B for further discussion).

## 4 Experiments

In this section we empirically validate the effectiveness of FIT across a range of popular LLMs of varying sizes and on different NLP datasets, including classification and multi-choice question-answering (MCQA) tasks.

Before reporting the main results, we introduce the evaluation metric (focus accuracy) that we report, baselines, models, and training settings used throughout the experiments. We first demonstrate in Section 4.1 that FIT generalises to subtle textual features and handles feature-value distribution shifts on the SMNLI dataset, a sub-sampled version of the MNLI dataset (Williams et al., 2018). In Appendix E, we additionally verify that FIT performs well on the SS dataset, a synthetic sentiment analysis dataset derived from SST-5 (Socher et al., 2013b). Finally, in Section 4.2, we show that FIT has practical, real-world impact by effectively mitigating bias in the BBQ dataset (Parrish et al., 2022), where we further illustrate FIT's ability to generalise to new features seen for the first time when performing inference.

Although the primary focus of FIT is on adaptively steering LLMs at inference time, which is what we focus on in this paper, we include an additional debiasing experi-

ment for comparison on the BBQ dataset in Appendix I for completeness. While bias mitigation is a valuable and natural application of FIT, it is not its primary objective. Instead, this inclusion highlights FIT's broader utility as a tool for model control and adaptability, demonstrating that it performs on par with dedicated bias mitigation techniques while also offering the unique advantage of test-time steerability.

**Metrics.** We define the *focus accuracy* for a focus instruction $I_{\text{focus}} \in \mathcal{I}_{\text{focus}}$ as the proportion of samples where the model's prediction aligns with the focus label, $y_{\text{focus}}$, as specified in Definition 1.

**Definition 3** (Focus Accuracy, $\mathcal{A}_{\text{focus}}$). For a sample $(x, y) \sim p_{\text{data}}$ and a fixed focus instruction $I_{\text{focus}} \in \mathcal{I}_{\text{focus}}$, we consider a model's prediction of the focus label $\hat{y} \sim p_\theta(\cdot | I, I_{\text{focus}}, x)$. Focus accuracy for focus instruction $I_{\text{focus}}$, denoted $\mathcal{A}_{\text{focus}}(I_{\text{focus}})$, is defined as

$$\mathcal{A}_{\text{focus}}(I_{\text{focus}}) = \frac{1}{|\mathcal{D}|} \sum_{(x,y) \in \mathcal{D}} \mathbf{1}(\hat{y} = y_{\text{focus}}), \qquad (5)$$

where $\mathbf{1}(\hat{y} = y_{\text{focus}})$ is the indicator function that equals 1 if the model's prediction $\hat{y}$ matches the focus label $y_{\text{focus}}$, and 0 otherwise.

We report focus accuracy for each model on all dataset splits, using the prompt types and focus instructions detailed in Appendix C. Generations are evaluated through simple pattern matching due to the use of constrained beam decoding (Anderson et al., 2017). See Appendix D.2 for further details.

**Models and Training Settings.** We evaluate FIT using three popular LLMs that span a range of model sizes: Llama-3.1-8B-Instruct (Dubey et al., 2024), Mistral-7B-Instruct-v0.3 (Jiang et al., 2023), and Vicuna-13B-v1.5 (Chiang et al., 2023). The models are fine-tuned using parameter-efficient SFT with LoRA (Hu et al., 2021), leveraging Hugging Face's SFTTrainer (Wolf et al., 2020). Early stopping is applied based on validation loss, as defined in Equation (3). For generation, we use constrained beam decoding (Anderson et al., 2017) and use fully verbalised (natural language) labels during both training and testing, except for the multi-choice BBQ dataset. Focus accuracies are reported over four independent repeats for each experiment. For further training details, refer to Appendix D.

**Baselines.** We compare against the following baselines in the main section of the paper: a few-shot baseline (Manikandan et al., 2023) and a SFT baseline. The SFT baseline, $\text{SFT}(y_{\text{focus}})$, follows the same setup as the FIT method (trained on sampled inputs and focus labels), but without the inclusion of focus instructions during training. This ensures a fair comparison between FIT and the baseline, as both methods are trained on the same examples and labels

(i.e., focus labels $y_{\text{focus}}$), with the only difference being the inclusion of focus instructions in FIT. This setup allows us to isolate and evaluate the specific impact of incorporating focus instructions.

Recent findings (Wu et al., 2025) indicate that LSMs significantly underperform compared to SFT methods. Hence, we include SFT baselines rather than LSM baselines within this work. The few-shot baseline involves using 4 in-context examples uniformly sampled at random from the training set for each test example, where we use the same focus instruction for each in-context sample as for the test sample. In Appendix D.4, we detail and include two additional baselines: zero-shot and vanilla SFT for a more complete comparison with FIT.

## 4.1 FIT Performs Well on the SMNLI Dataset and Generalises Under Distribution Shift

We evaluate our method on a dataset with subtle textual features. Specifically, we construct an NLI dataset by sub-sampling from MNLI (Williams et al., 2018), where we induce a spurious correlation between text genres and labels through our subsampling process. We call this subsampled dataset the SMNLI dataset.

**Dataset Construction**. Figure 4 illustrates the data generating process (DGP) describing how we subsample examples to induce spurious correlations between feature value $s \in \text{Val}(S)$ and a particular associated label $y_s \in \text{Val}(Y)$. The feature set that we consider is defined as $\mathcal{F} = \{C, S\}$, where $C$ is the NLI relationship and $S$ is the genre of a given premise-hypothesis pair.

The co-occurrence rate of genres and their spuriously associated labels is governed by $\rho_{\text{spurious}}$, which varies across the test sets discussed in Section 3. We ensure that $\rho_{\text{spurious}}$ is the same for all feature values in $\text{Val}(S)$ within each dataset split. In particular, we set $\rho_{\text{spurious}}$ to be $1/3, 1/3, 0.9, 0.1$ and $0.9$ on $\mathcal{D}_{\text{train}}, \mathcal{D}_{\text{iid}}, \mathcal{D}_{\text{high}}, \mathcal{D}_{\text{low}}$ and $\mathcal{D}_{\text{flipped}}$ respectively. Moreover, we hold out specific genres at test time to evaluate our model's ability to generalise under distribution shift when feature values change. We do this by sub-sampling a held-out portion of the MNLI dataset. During training, we use three selected genres $\mathcal{S} = \{\text{government, fiction, travel}\}$ to train and evaluate our models. We additionally add three held-out genres $\mathcal{S} = \{\text{facetoface, nineeleven, verbatim}\}$. We again ensure that $\rho_{\text{spurious}}$ is constant within each of these shifted splits across feature values, and use the same set of corresponding $\rho_{\text{spurious}}$ as within the SMNLI test sets described above. This process is again governed by a DGP shown in Figure 4, giving us precise control over the data synthesis process for the SMNLI dataset. Further details of the SMNLI dataset can be found in Appendix F.

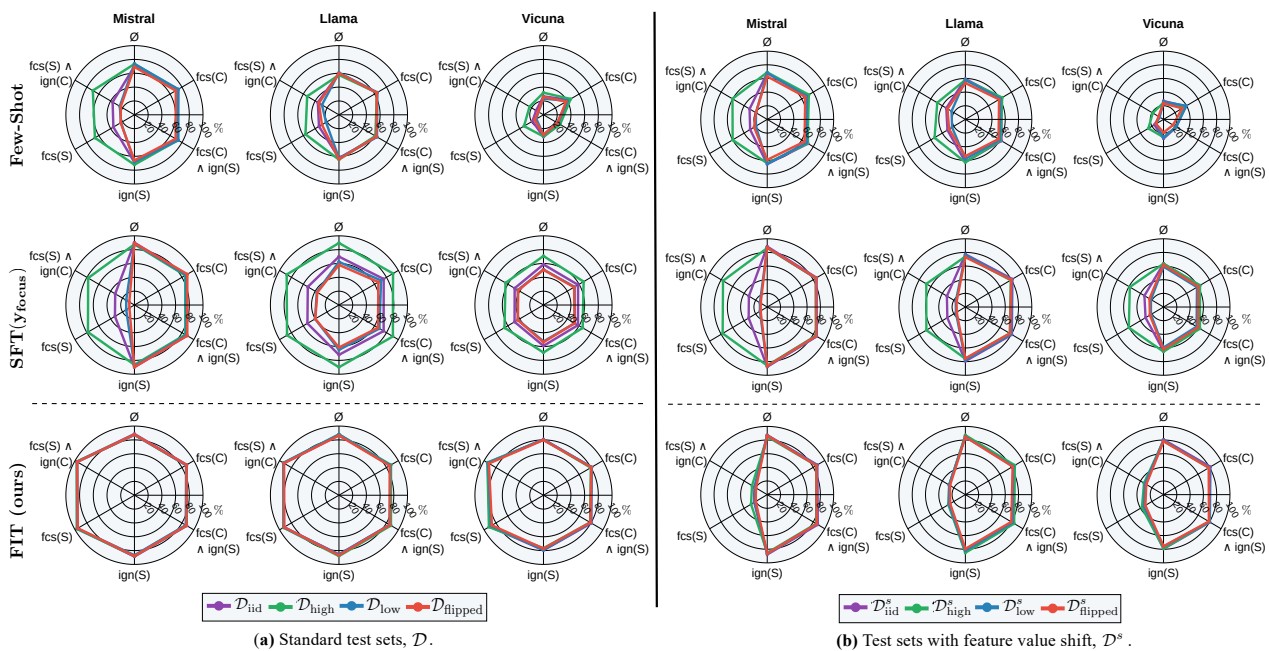

(a) Standard test sets, $\mathcal{D}$.

(b) Test sets with feature value shift, $\mathcal{D}^s$.

*Figure 3.* **SMNLI Focus Accuracies** ($\uparrow$). Mean focus accuracy ($\mathcal{A}_{\text{focus}}$) of baselines and FIT models on the (a) SMNLI standard test sets $\mathcal{D}$, and (b) SMNLI test sets under feature value shift $\mathcal{D}^s$. The maximum standard deviations of FIT, SFT($y_{\text{focus}}$) and few-shot methods across models and $\mathcal{I}_{\text{focus}}$ are 6.47%, 7.98% and 0.500% respectively. fcs = focus, ign = ignore.

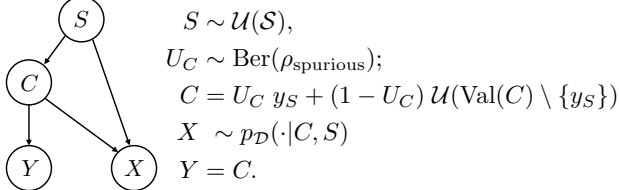

$$S \sim \mathcal{U}(\mathcal{S}),$$
$$U_C \sim \text{Ber}(\rho_{\text{spurious}});$$
$$C = U_C \, y_S + (1 - U_C) \, \mathcal{U}(\text{Val}(C) \setminus \{y_S\})$$
$$X \sim p_{\mathcal{D}}(\cdot | C, S)$$
$$Y = C.$$

*Figure 4.* **SMNLI DGP**. DGP describing the subsampling process of MNLI to introduce the spurious genre feature $S$. Here, $\mathcal{U}(\mathcal{S})$ is the uniform distribution over genres, $\text{Val}(C) = \{0, 1, 2\}$ are the NLI labels (with $y_S$ tied to each $S$), and $p_{\mathcal{D}}(\cdot | C, S)$ is the MNLI conditional distribution over premise–hypothesis pairs.

**Results.** Figure 3 (a) depicts the focus accuracy results of the three models on the SMNLI test splits. We observe that for both the core and genre feature, FIT achieves very high focus accuracy, significantly improving over the baselines. This demonstrates that FIT effectively trains the model to handle subtle textual features, allowing it to dynamically focus on or disregard these features when making predictions.

Figure 3 (b) shows the focus accuracy of models on the feature-shifted test sets. When focusing on the core feature or ignoring the spurious feature, the model maintains strong performance in terms of focus accuracy, even on unseen genre values (over 80% focus accuracy for FIT models on the third row of Figure 3 (b)), generalising generally more robustly than the baseline methods when focus is over core features.

While we observe low focus accuracy when focusing on

spurious features, this is expected because the spurious labels associated with these new genres were not encountered during training, so the model cannot know these new relations. Importantly, FIT remains steerable, changing its predictions depending on what is focused on, and this holds steady across all predictivity levels for the new spurious genres. In contrast, the baselines show decreasing focus accuracy as predictivity decreases, indicating a tendency to predict the causal label under distribution shift. This shows that these models do not change their behaviour when instructed to change their focus, and thus have poorer steering ability under distribution shift compared to FIT.

> ***Key Takeaways.*** FIT achieves strong steerability, which is maintained under distribution shift. This demonstrates FIT's generalisation to new contexts with changing feature values.

### 4.2 FIT Steers Behaviour in the Presence of Social Bias Data and Generalises to Unseen Features

**Bias Benchmark for QA (BBQ) Dataset**. We experiment with BBQ (Parrish et al., 2022), a MCQA benchmark annotated with social biases that are relevant to any given answer, such as stereotypes that would imply a given answer to an otherwise ambiguous question (see Figure 1).

We consider the following feature set $\mathcal{F} = \{\text{question context, gender identity, race/ethnicity, ...}\}$, which contains one core feature (question context used to

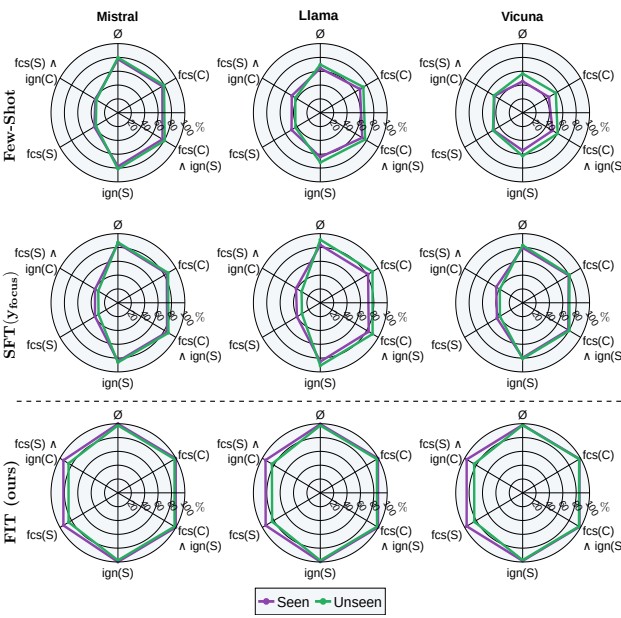

*Figure 5.* **BBQ Focus Accuracies** (↑). Mean focus accuracy ($\mathcal{A}_{focus}$) of baselines and FIT on the BBQ dataset. The maximum standard deviations of FIT, SFT($y_{focus}$) and few-shot methods across models and $\mathcal{I}_{focus}$ are 4.07%, 10.7% and 0.600% respectively. fcs = focus, ign = ignore.

answer a posed question) and 9 bias features. Of the 9 bias features, we focus-tune models with respect to 6, and test on these 6 features plus the remaining 3 bias features in order to test how well FIT generalises to features that are not seen during focus tuning. Here, we consider the spurious features to be the presence of a particular social group (e.g., men or women) in the question context, and spurious answers to be those that would be indicated by relying on social stereotypes rather than the specific question context (e.g., see Figure 1). The stereotyped response used to determine spurious answers for these bias features are provided as part of the BBQ dataset.

**Results.** Figure 5 shows the focus accuracy results of the three models on the BBQ dataset, visualising performance on features seen during training and unseen, held-out features. The models demonstrate high and comparable focus accuracy across both seen and unseen bias features, indicating that FIT generalises well to unseen features, including nuanced reasoning about group stereotypes. This highlights the usefulness of FIT in mitigating social biases in LLM responses. Specifically, FIT can effectively learn, reason about, and rule out biases when formulating responses, making it a practical tool for bias mitigation.

> ***Key Takeaways.*** FIT effectively teaches models to adjust their responses based on knowledge of social biases. This generalises to biases not seen during training, indicating FIT's utility for bias mitigation.

## 5 Ablation Studies

### 5.1 Extending FIT to NLG Tasks.

The primary focus of our experiments has been on classification and MCQA datasets, due to the cost and difficulty of collecting high-quality natural language generation (NLG) benchmarks. As an initial step towards extending FIT to NLG tasks, we introduce BBQ-NLG, a dataset that follows the BBQ MCQA setup (see Figure 5) except where we now drop the fixed answer options for examples and require the model both to identify all plausible answers from context and to generate the correct answer in a fully verbalised form. To assess generation accuracy against ground-truth responses, we use a pre-trained Llama-3.1-8B-Instruct model as an automated judge. Further details and full results are given in Appendix H

The results shown in Figure 6 confirm that FIT can steer models effectively at inference time and generalise to novel, unseen features even in this NLG-style setting, underscoring that extending FIT to NLG tasks is a particularly promising avenue for future research.

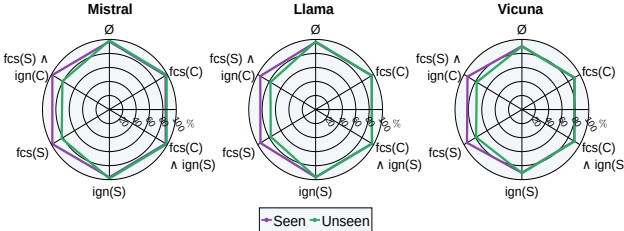

*Figure 6.* **BBQ-NLG FIT Focus Accuracies** (↑). Mean focus accuracy ($\mathcal{A}_{focus}$) of FIT models on the BBQ-NLG dataset. The maximum standard deviation across across FIT models and $\mathcal{I}_{focus}$ is . fcs = focus, ign = ignore.

### 5.2 Robustness to Prompt Phrasing.

Instruction-tuned models can overfit to the exact wording of their prompts, faltering on paraphrases (Ghosh et al., 2024). Figure 7 contrasts SMNLI focus accuracy with test-time focus instructions $\mathcal{I}_{focus}$ are either (i) the original training prompts given in Figure 9 or (ii) one of ten ChatGPT-paraphrased variants for each instruction type in Equation (2). Across splits and focus features, focus accuracy varies negligibly, indicating that FIT is robust to the particular phrasing of focus instructions.

### 5.3 FIT does not Affect General Capabilities.

Prior work has shown that SFT can erode the instruction-following capabilities of LLMs (Fu et al., 2024; Dou et al., 2024). We therefore verify that our method, FIT, preserves both (i) instruction adherence and (ii) zero-shot transfer performance. All FIT models in this section are trained only on the SMNLI dataset (see Section 4.1).

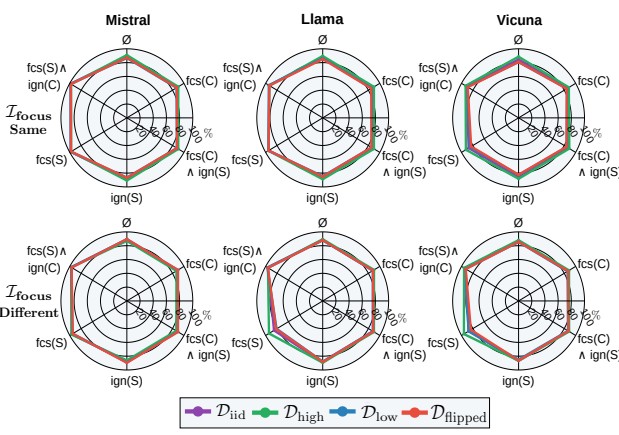

*Figure 7.* **Different Training and Test $\mathcal{I}_{\textbf{focus}}$ Focus Accuracy (↑).** SMNLI focus accuracies ($\mathcal{A}_{\text{focus}}$) when test focus instructions $I_{\text{focus}}$ prompts are drawn from the training focus instruction set (top) (see Figure 9) versus a paraphrased focus instruction set (bottom). fcs = focus, ign = ignore.

**Instruction Following (Alpaca-GPT Dataset).** We sample 500 prompts from the Alpaca-GPT set (Peng et al., 2023) and score each model's response with GPT-4o (Achiam et al., 2023) on a 1–5 scale (5 = perfect alignment). Table 1 reports the mean score before and after FIT along with two-sided Wilcoxon signed-rank $p$-values. Across Llama, Mistral and Vicuna, the differences in ratings are small ($\leq 0.05$ points) and never significant ($p > 0.05$), confirming that FIT preserves instruction-following capabilities.

| Model | Llama | Mistral | Vicuna |
|---|---|---|---|
| Pre-Trained Avg. Rating (↑) | 3.51 | 3.65 | 3.46 |
| FIT Avg. Rating (↑) | 3.45 | 3.65 | 3.50 |
| $p$-value | **0.57**$_{>0.05}$ | **0.81**$_{>0.05}$ | **0.41**$_{>0.05}$ |

*Table 1.* **Instruction Following After FIT.** For each base model (columns), we report the pre-trained and FIT average GPT-4o ratings, and the two-sided Wilcoxon Signed-Rank $p$-value testing the difference between the distributions of ratings.

**Zero-Shot Transfer (MMLU).** We next measure zero-shot transfer to MMLU (Hendrycks et al., 2021), a MCQA dataset, testing problem solving and general world knowledge of models. Table 2 shows accuracy and perplexity (across entire model vocabulary) for pre-trained and FIT Llama and Mistral models. FIT changes accuracy by at most 0.8%, while showing lowering perplexity, indicating that task performance of pre-trained models has not been sacrificed through FIT. This demonstrates that FIT does not hurt existing transfer performance of base models in a zero-shot setting.

### 5.4 Model Size Ablation

We further assess FIT across the Qwen-2.5-Instruct (Yang et al., 2024a) family of models on three scales (1.5B, 3B

| Model | Llama | | Mistral | |
|---|---|---|---|---|
| | Pre-Trained | FIT | Pre-Trained | FIT |
| Accuracy (↑) | 30.4 | 29.6 | 29.4 | 29.0 |
| Perplexity (↓) | 6.29 | 2.79 | 15.2 | 5.22 |

*Table 2.* **Zero-Shot MMLU After FIT.** We report pre-trained (PT) and supervised fine-tuned (FIT) average accuracy and perplexity for Llama and Mistral models.

and 7B parameters) on the BBQ dataset under the same training conditions as in Section 4.2. As shown in Figure 8, FIT shows strong steerability across all model sizes, even for the smallest 1.5B model. Moreover, we see that performance generally scales with model size. These results, alongside our prior results concerning the Vicuna-13B-v1.5 model demonstrate FIT's robustness to model capacity and its favourable scaling behaviour.

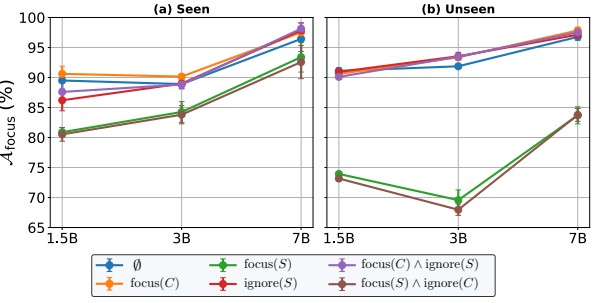

*Figure 8.* **Model Size Ablation.** Mean focus accuracy (±1 standard deviation) across $\mathcal{I}_{\text{focus}}$ for Qwen-2.5-Instruct models at 1.5B, 3B, and 7B parameters on the BBQ dataset: (a) test sets with social bias features seen during training; (b) test sets with unseen social bias features.

## 6 Conclusion

In this work, we introduce Focus Instruction Tuning (FIT), a method designed to steer the behaviour of LLMs by focusing on or ignoring specific features when formulating responses. Across a range of tasks and settings, we demonstrate that FIT provides dynamic and precise control over LLM behaviour at inference time, enabling users to adapt model responses base on natural language instructions. This steering behavoi holds even in the context of distribution shifts over feature values or when generalising to unseen features. Furthermore, our approach can address challenges such as mitigating the influence of known stereotypes that might otherwise impact responses, showcasing one of its many applications. Thus, FIT represents a step toward enabling more robust, steerable, fair, and controllable LLMs.

## Acknowledgments

This work is supported by the UKRI grant: Turing AI Fellowship EP/W002981/1. Alasdair Paren is supported by UK ASIS Systemic AI Safety Grant. Francesco Pinto ac-

knowledges support through the DSI, University of Chicago. Adam Davies is supported by the National Science Foundation and the Institute of Education Sciences, U.S. Department of Education, through Award #2229612 (National AI Institute for Inclusive Intelligent Technologies for Education). Any opinions, findings, and conclusions or recommendations expressed in this material are those of the author(s) and do not necessarily reflect the views of National Science Foundation or the U.S. Department of Education.

## Impact Statement

The ability to dynamically steer model behaviour by focusing on or ignoring features, as enabled by FIT, holds significant potential for reducing algorithmic discrimination and mitigating harms. Practitioners can leverage FIT to identify and correct biases by measuring discrepancies in behaviour when a model focuses on or ignores specific features. Additionally, FIT enhances explainability by attributing model predictions to input features, enabling more transparent and productive human-AI collaboration. This supports ethical and responsible decision-making by assessing whether predictions are justified. FIT also enhances robustness by prioritising stable core features expected to generalise across domains while ignoring spurious, domain-specific biases, making it a valuable tool for fairness, explainability, and robustness. However, risks include potential misuse by bad actors to bias models, though this is not unique to FIT and could already be achieved through biased fine-tuning.

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

# A Author Contributions

- **Tom A. Lamb:**

  ○ *Idea:* Turned the initial project ideas and directions into the concrete methodology described in Section 3.

  ○ *Experiments and Methods:* Designed and carried out all experiments in the paper. These are described in Figure 3, Figure 5, Appendix E and Section 5.

  ○ *Dataset Creation:* Designed and carried out the synthesis of the SS and SMNLI dataset presented in Appendix E and Section 4.1.

  ○ *Analysis:* Led the analysis of all empirical results.

  ○ *Paper Writing:* Led writing of all sections (main paper and appendix), including both the initial drafts of the paper and the final version of the paper presented here; and led literature review on spurious feature learning, instruction-tuning and helped with the aligning LLMs section of Section 2.

- **Adam Davies:**

  ○ *Idea:* Co-conceived the research problem and core approach of FIT (with Francesco Pinto).

  ○ *Supervision:* Led advising of methodology and experimental design.

  ○ *Analysis:* Advised and assisted with analysis of all empirical results.

  ○ *Paper Writing:* Co-wrote drafts of Abstract, Section 1 and Section 2; led literature review for mechanistic interpretability, aligning LLMs, and latent steering in Section 2, and contributed to literature review for spurious feature learning and instruction tuning; and helped edit and revise all sections of the paper.

- **Alasdair Paren:**

  ○ *Supervision:* Co-advised methodology and experimental design with a particular focus on the simplicity bias and shortcut learning.

  ○ *Paper Writing:* Provided feedback on drafts.

- **Philip Torr:**

  ○ *Supervision:* Provided general advice on initial project direction; helped secure compute resources for the experiments conducted in this paper.

  ○ *Paper Writing:* Provided draft feedback on the initial sections of the paper.

- **Francesco Pinto:**

  ○ *Idea:* Co-conceived the research problem and core approach of FIT (with Adam Davies).

  ○ *Supervision:* Co-advised methodology and experimental design.

  ○ *Paper Writing:* Provided feedback on earlier versions of the drafts of the paper.

# B Limitations and Future Work

**Requirement for Annotated Spurious Features.** While FIT relies on prior identification of spurious features and their focus labels, this requirement does not limit its practical applicability. Instead, it reflects standard industry and research practices for constructing transparent and reliable models. Below, we clarify how FIT remains adaptive and versatile even when feature annotation is partial or evolving:

- *Alignment with Established Practices:* FIT's reliance on pre-identified spurious features aligns with widely adopted industry and research norms (OpenAI, 2024; Microsoft, 2020). Identifying potential spurious features and confounders in datasets is a foundational step in achieving robust machine learning systems. This process ensures that both training and validation phases are informed by an understanding of data correlations, minimising the risk of deploying models with unknown biases.

- *Regulatory and Ethical Expectations:* Regulatory frameworks and ethical guidelines increasingly require the explicit identification and mitigation of problematic features (of the European Parliament, 2016). Corresponding initiatives aim to define and enforce measurable categories of "violating behaviour" in AI models. By providing a mechanism to steer model behaviour based on these identified features, FIT effectively complements efforts to promote fair and transparent predictions (Guldimann et al., 2024; Zeng et al., 2024).

- *Post-Deployment Mitigation:* Despite careful pre-deployment analysis, spurious features or correlations may only become apparent once a model is in active use. FIT accommodates this by allowing developers to incorporate newly identified spurious features via updated focus instructions, enabling rapid iterative refinement without retraining from scratch. This adaptability ensures continuous improvement, even in highly dynamic environments.

- *FIT's Versatility Without Exhaustive Pre-Identification:* Crucially, FIT does not require an exhaustive list of spurious features to be effective. For instance, a user can provide focus instructions such as "focus on casual" without enumerating every possible irrelevant attribute in the dataset. This flexibility expands FIT's applicability to scenarios where feature annotation is incomplete or ongoing.

- *Compatibility with Automated Spurious Feature Identification:* FIT also works seamlessly with automated methods for detecting spurious features (Wang & Culotta, 2020; Wang et al., 2022; Zhou et al., 2024b; Zheng et al., 2024). Whether spurious features are labelled manually or derived from automated detection, they can be harnessed by FIT's focus instructions at inference time. This compatibility enables a comprehensive approach to managing known issues and responding to newly uncovered features as they arise.

In summary, annotating spurious features beforehand is not a strict limitation. FIT can be flexibly applied, allowing model behaviour to evolve in tandem with new feature discoveries or changing requirements, making it a broadly applicable technique for steering model outputs based on both prior knowledge and ongoing insights.

**Scope of Experiments and Extensions to Open-Ended Tasks.** Our experiments primarily focus on classification and multiple-choice QA datasets due to the cost and challenges associated with curating high-quality datasets for open-ended NLG tasks. However, this reflects a pragmatic prioritisation of introducing a novel methodology over exhaustive data collection, rather than a limitation of FIT itself. Whilst we provide positive evidence through our BBQ-NLG ablation in Section 5.1, extending FIT to open-ended tasks such as summarisation or translation, remains an key direction for future research, as does exploring its ability to generalise across diverse task categories using setups similar to FLAN (Longpre et al., 2023).

**Overlapping Features and Ambiguities.** Additionally, our evaluation on the HANS dataset Appendix K revealed challenges when addressing overlapping or less-distinctive features. While FIT demonstrated strong performance in generalising and steering models based on identified features, overlapping heuristics can introduce ambiguity, highlighting the need for further refinements in handling such cases. Despite these limitations, FIT represents a promising foundation for enabling more robust, fair, and controllable LLMs across a range of tasks.

**Handling Multiple Features.** While our current experiments provide initial evidence that FIT can handle combined instructions, specifying both a feature to focus on and a feature to ignore, we recognise the importance of scaling this capability. An important direction for future research is extending FIT to manage instructions involving multiple features simultaneously, including cases where models are required to focus on or ignore several attributes in tandem.

# C   FIT Focus Instructions and Prompt Templates

**Prompt Templates.** In Figure 10 we provide the prompt templates that we use for FIT training and for general evaluation across all methods. Note, that we drop the feature considerations section and focus instruction from the prompt to form the default prompt $\emptyset$ introduced in Equation (2).

**Focus Instructions.** In Figure 9 we show the focus instruction formats for the different focus instructions introduced in Equation (2), defining $\mathcal{I}_{\text{focus}}$ throughout our experiments.

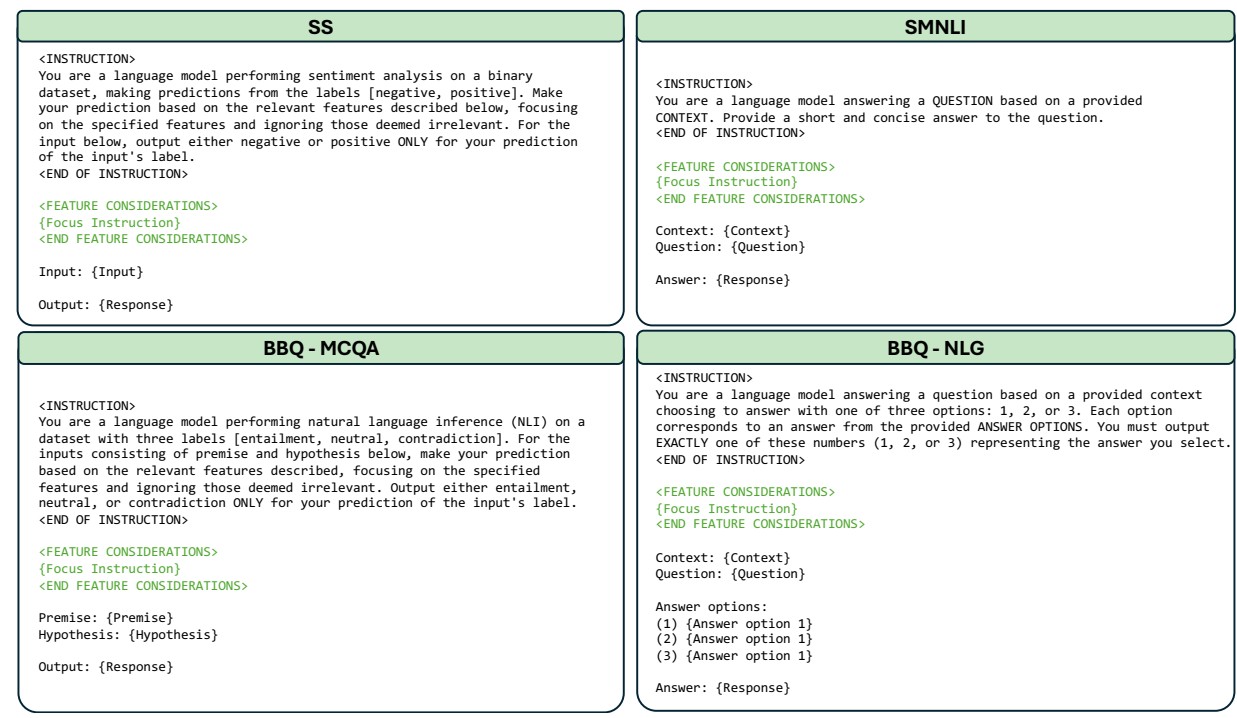

*Figure 9.* **Focus Instructions.** Focus instructions that are used for focussing and ignoring features $F_i, F_j \in \mathcal{F}$ during FIT training and for general evaluation.

```
SS

<INSTRUCTION>
You are a language model performing sentiment analysis on a binary
dataset, making predictions from the labels [negative, positive]. Make
your prediction based on the relevant features described below, focusing
on the specified features and ignoring those deemed irrelevant. For the
input below, output either negative or positive ONLY for your prediction
of the input's label.
<END OF INSTRUCTION>

<FEATURE CONSIDERATIONS>
{Focus Instruction}
<END FEATURE CONSIDERATIONS>

Input: {Input}

Output: {Response}
```

```
SMNLI

<INSTRUCTION>
You are a language model answering a QUESTION based on a provided
CONTEXT. Provide a short and concise answer to the question.
<END OF INSTRUCTION>

<FEATURE CONSIDERATIONS>
{Focus Instruction}
<END FEATURE CONSIDERATIONS>

Context: {Context}
Question: {Question}

Answer: {Response}
```

```
BBQ - MCQA

<INSTRUCTION>
You are a language model performing natural language inference (NLI) on a
dataset with three labels [entailment, neutral, contradiction]. For the
inputs consisting of premise and hypothesis below, make your prediction
based on the relevant features described, focusing on the specified
features and ignoring those deemed irrelevant. Output either entailment,
neutral, or contradiction ONLY for your prediction of the input's label.
<END OF INSTRUCTION>

<FEATURE CONSIDERATIONS>
{Focus Instruction}
<END FEATURE CONSIDERATIONS>

Premise: {Premise}
Hypothesis: {Hypothesis}

Output: {Response}
```

```
BBQ - NLG

<INSTRUCTION>
You are a language model answering a question based on a provided context
choosing to answer with one of three options: 1, 2, or 3. Each option
corresponds to an answer from the provided ANSWER OPTIONS. You must output
EXACTLY one of these numbers (1, 2, or 3) representing the answer you select.
<END OF INSTRUCTION>

<FEATURE CONSIDERATIONS>
{Focus Instruction}
<END FEATURE CONSIDERATIONS>

Context: {Context}
Question: {Question}

Answer options:
(1) {Answer option 1}
(2) {Answer option 1}
(3) {Answer option 1}

Answer: {Response}
```

*Figure 10.* **FIT Prompt Templates.** Prompt templates used for FIT training and evaluation across all four datasets that we investigate over.

# D  FIT Training, Optimisation and Evaluation

## D.1  FIT Training and Optimisation

**FT Optimisation.** Algorithm 1 gives precise details on how we implement FIT in practice when performing ERM of a model using the FIT training objective given in Equation (3) on a given training set.

---

**Algorithm 1** Algorithm for Focus Instruction Tuning  (FIT) Training Procedure to Optimise Equation (3).

---

1: **Input:** Dataset $\mathcal{D} = \{(x_i, y_i)\}_{i=1}^N$, The feature set $\mathcal{F}$, focus instructions $\mathcal{I}_{\text{focus}}$, instruction $I$, model parameters $\theta$, batch size $B$, number of epochs $E$, step size $\eta$, and focus label mapping $y_{\text{focus}} = y_{\text{focus}}(I_{\text{focus}}, y, s)$.
2: **Initialise:** Model parameters $\theta$, optimiser.
3: **for** epoch $= 1$ to $E$ **do**
4:     **for** mini-batch $\{(x^b, y^b)\}_{b=1}^B$ from $\mathcal{D}$ **do**
5:         **for** each $(x^b, y^b)$ in the mini-batch **do**
6:             Identify spurious feature value $s^b$ in $x^b$.
7:             Sample focus instruction $I_{\text{focus}}^b \sim p_{\mathcal{I}_{\text{focus}}}$.
8:             Compute $y_{\text{focus}}^b = y_{\text{focus}}(I_{\text{focus}}^b, s^b, y^b)$ .
9:         **end for**
10:         Compute average loss given through empirical estimator of the loss defined in Equation (3) over the batch:

$$\ell(\theta) = \frac{1}{B} \sum_{b=1}^B -\log p_\theta(y_{\text{focus}}^b | I, I_{\text{focus}}^b, x^b).$$

      .
11:         Update model parameters $\theta$ using optimiser:

$$\theta \leftarrow \theta - \eta \nabla_\theta \ell(\theta).$$

12:     **end for**
13: **end for**
14: **Output:** Optimised model parameters $\theta$ .

---

**FT Training Settings.** We use LoRA (Hu et al., 2021) for parameter-efficient fine-tuning. We target the query and value projection matrices within each LLM and use LoRA $r = 16$ and $\alpha = 32$ across models.

We implement early stopping on a held-out validation set based on the cross-entropy loss over focus labels $y_{\text{focus}}$ corresponding to randomly sampled focus instructions - this matches the context in which the models will be evaluated. We obtain this set by splitting our training sets in a 90/10% ratio for training and validation splits respectively. We use a patience of 4 validation evaluation steps, which occur after a fixed number of steps.

During training, we define $p(\mathcal{I}_{\text{focus}})$ by placing a small probability (in our experiments, 0.05) on the empty focus instruction $\emptyset$. We then uniformly distribute the remaining probability mass over the non-empty focus instructions.

**Choice of $\rho_{\text{spurious}}$ During Training.** In our synthetic experiments described in Figure 3 and Appendix E, we set up a controlled environment by imposing two independence conditions: $Y \perp\!\!\!\perp S$ and $Y_S \perp\!\!\!\perp C$. These ensure that (i) the ground-truth label cannot be predicted using the spurious feature $S$, and (ii) the spurious label cannot be predicted using the core feature $C$. By removing direct correlations between these features and labels, the model learns to focus on the specified feature, without being influenced by another feature, avoiding any potential shortcuts that could be exploited if these conditions did not hold.

- **Independence $Y \perp\!\!\!\perp S$:** This condition prevents the model from leveraging spurious feature $S$ to predict ground-truth label $Y$. With no predictive signal from $S$ to $Y$, the model must rely exclusively on the core feature $C$ for accurate label predictions. This design choice safeguards the model from overfitting to spurious correlations, thereby maintaining robust performance under distribution shifts. Moreover, removing any inherent relationship between $S$ and $Y$ ensures that for focus instruction intending for the model to utilise the core feature $C$ only during inference, the model cannot exploit a potential shortcut using $S$; it must utilise the core feature alone for prediction in this scenario. This enforces prediction only through the specified feature indicated through the focus instruction passed to the model.

- **Independence** $Y_S \perp\!\!\!\perp C$: This condition serves a complementary role in preventing the model from exploiting the core feature $C$ when predicting spurious labels $Y_S$. By ensuring $C$ carries no information about $Y_S$, the model cannot use the true task feature $C$ as a shortcut for spurious-label predictions; it must again learn to only use the specified feature within the passed focus instructions for making predictions.

While these conditions represent an ideal setting, they are not strictly necessary for FIT to work in practice. Indeed, real-world data rarely satisfies such independence assumptions. We illustrate the robustness of the method in more realistic scenarios through our BBQ experiments in Section 4.2 where correlations between $Y$ and $S$ or between $C$ and $Y_S$ may exist, as no subsampling or dataset manipulations have been made to enforce these conditions. By examining both controlled environments and more natural datasets, we demonstrate the robustness of FIT.

To achieve the aforementioned independence assumptions in our synthetic SS and SMNLI datasets, we set $\rho_{\text{spurious}} = 1/N$, where $N$ is the number of class labels. Additionally, we enforce a balanced label distribution in the training set to eliminate any indirect biases that could correlate $S$ with $Y$. As shown in Appendix E and Appendix F, these conditions are sufficient to guarantee $Y \perp\!\!\!\perp S$ in the training data, enabling the model to effectively learn steerable behaviour from focus instructions.

## D.2 Evaluation

**Generation Settings.** We generate responses from models using constrained beam-decoding (Anderson et al., 2017) with $4$ beams. This ensures that the answer labels for each classification task that we investigate appear in the model's output. We limit the maximum number of newly generated tokens to be $5$ to stop any unnecessary text given after the model's initial classification prediction.

**Computing the Focus Accuracy Metric.** We report the focus accuracy $\mathcal{A}_{\text{focus}}$ of generations when evaluating models. As we are guaranteed to include the task labels within the model's response through constrained decoding, and have reduced the maximum number of tokens that a model can generate at inference-time, we simply check to see if the focus label, $y_{\text{focus}}$, is within the model's response or not in order to determine if the model's response is correct or not.

## D.3 Dataset Sizes

The sizes of each of the splits of the SS, SMNLI and BBQ datasets are given in Table 3.

*Table 3.* **Dataset Sizes.** Dataset split sizes for SS, SMNLI and BBQ datasets.

|            | SS   | SMNLI | BBQ   |
|------------|------|-------|-------|
| Training   | 5296 | 7200  | 16700 |
| Validation | 1324 | 1800  | 1590  |
| Test       | 1818 | 900   | 2352  |

## D.4 Complete Set of Baselines

In addition to the baselines that we present in the main paper, namely few-shot and SFT($y_{\text{focus}}$), we include two additional baselines to further supplement these results. We give the complete list of baselines below:

**Zero-Shot Baseline.** We include a zero-shot inference baseline using the original pre-trained models without additional fine-tuning on any dataset. No in-context examples are used at inference time. The model is tested on the full set of focus instructions prompts detailed in Equation (2).

**Few-Shot Baseline.** This second baseline compares FIT training to few-shot inference, using the original pre-trained models without additional fine-tuning on our spurious datasets. Specifically, we use 4 in-context examples across all datasets. For the in-context examples, we concatenate multiple examples one after the other. Each in-context example contains the same focus instruction as the test example for which they serve as context. The model is tested on the full set of focus instructions prompts detailed in Equation (2).

**SFT($y_{\text{focus}}$) Baseline.** We implement an SFT baseline that follows the same training procedure as FIT, except during training, we exclude any focus instructions from the input prompts while still training on the focus labels. This provides a fair comparison with FIT, as models are trained on the same input text-label pairs. The rest of the training setup, including

hyperparameters and early stopping, remains identical to the FIT training setup. The model is tested on the full set of focus instructions prompts detailed in Equation (2).

**SFT($y$) Baseline.** We implement a vanilla SFT baseline that simply trains a model using SFT on inputs and their ground truth labels (as opposed to focus labels in the SFT($y_{\text{focus}}$) baseline). During training, only standard IT prompts are used corresponding to the default prompt $\emptyset$, with no additional focus instructions included. The rest of the training setup, including hyperparameters and early stopping, remains identical to the FIT training setup. The model is tested on the full set of focus instructions prompts detailed in Equation (2).

We give the full set of results for all datasets and models across the complete set of baselines listed above in Figure 12, Figure 14, and Figure 15.

# E    Spurious Sentiment (SS) Dataset

We take a pre-existing dataset, in this case SST-5 (Socher et al., 2013a), and modify it in order to include a known spurious feature and create a spurious binary sentiment analysis dataset that we call the *spurious sentiment (SS) dataset*.

## E.1    Data-generating process (DGP)

We frame our DGP using a graphical model to describe the synthetic dataset that we create. We follow a similar model to that described in (Arjovsky et al., 2019), specifically the model used for generating their coloured MNIST dataset. We use the following variables within our graphical model:

- $C$ - true underlying sentiment, the core feature within this task, sampled from the original dataset.
- $\tilde{S}$ - proposed spurious feature sample, here this is the presence of the keywords "Pineapple" or "Bayesian". We represent this as a binary categorical variable with $\text{Val}(\tilde{S}) = \{\text{Pineapple}, \text{Bayesian}\}$. We note that, this restricts us to consider only one keyword appearing in a text at any given time.
- $S$ - the final included spurious feature that is naturally inserted using a LLM into the final SS dataset example $X$. The feature $S$ is a randomly flipped version of the proposed spurious feature $\tilde{S}$. So here $\text{Val}(S) = \{\text{Pineapple}, \text{Bayesian}\}$ also.
- $\tilde{X}$ - is a sampled example from the original SST-5 dataset.
- $X$ - original example $\tilde{X}$ but naturally augmented to include the spurious feature $S$, without changing the underlying sentiment of the example.
- $Y$ - final label for element $X$.

The graphical model describing the DGP of the SS dataset is given in Figure 11a. This admits a functional representation in the form:

$$C = f_C(U_C); \tag{6}$$
$$\tilde{X} = f(C, U_{\tilde{X}}); \tag{7}$$
$$\tilde{S} = f_{\tilde{S}}(C, U_{\tilde{S}}); \tag{8}$$
$$S = f_S(\tilde{S}, U_S); \tag{9}$$
$$X = f_X(\tilde{X}, S, U_X); \tag{10}$$
$$Y = f_Y(C, U_Y). \tag{11}$$

where $U_{(\cdot)}$ are variables introducing sources of randomness into the generating process. More explicitly, we consider the following set of equations, where $\mathcal{D}$ denotes the underlying dataset that we are manipulating:

$$C \sim \text{Ber}(\rho_C), \text{ where } \rho_C = \rho_C(\mathcal{D}); \tag{12}$$
$$\tilde{X} \sim p_{\mathcal{D}}(\cdot|C) ; \tag{13}$$
$$\tilde{S} = \begin{cases} Pineapple & \text{if } C = 0 \\ Bayesian & \text{if } C = 1 \end{cases}; \tag{14}$$
$$U_S \sim \text{Ber}(\rho_{\text{spurious}}); \tag{15}$$
$$S = \begin{cases} \tilde{S} & \text{if } U_S = 1 \\ \text{Val}(\tilde{S}) \setminus \tilde{S} & \text{if } U_S = 0 \end{cases}; \tag{16}$$
$$X = \text{LLM}(\tilde{X}, S); \tag{17}$$
$$Y = C, \tag{18}$$

Here, Ber denotes a Bernoulli distribution, and the function LLM denotes a LLM that naturally injects a spurious feature $S$ into example $\tilde{X}$ without altering its underlying sentiment. The variable $\rho_C$ governs the distribution of sentiment labels in the original binarised SST-5 dataset. Moreover, $p_{\mathcal{D}}(\cdot|C)$ denotes the conditional dataset distribution of the different input texts given $C$ (here we assume that we are just uniformly sampling text with the given sentiment $C$).

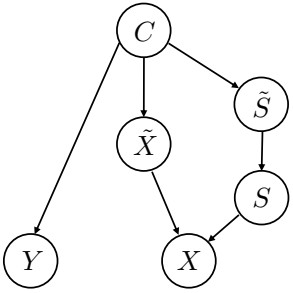

(a) **SS DGP**. Graphical model showing the data generating process for modifying examples from the SS dataset to introduce a new spurious keyword feature, $S$.

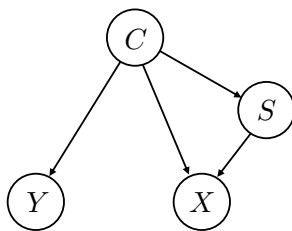

(b) **SS Causal Graph.** Causal graph showing showing the spurious correlation present between the keyword feature $S$ and the label $Y$ for examples in the SS dataset, induced through the described data augmentation process.

*Figure 11.* **SS DGP and Causal Graph.** The DGP and associated causal graph describing the generation of dataset examples and showing the causal structure within examples.

Through the above DGP, we introduce a new spurious feature within the dataset, $S$. Recalling that $S =$ Pineapple and $S =$ Bayesian correspond to the insertion of the keywords Pineapple and Bayesian respectively, we introduce the following spurious correlations between the final included feature values of $S$ and label $Y$:

(a) The presence of the word Pineapple in the text $X$, that is $S =$ Pineapple, is spuriously correlated with the label 0 (negative sentiment). Therefore, the spurious label associated with $S =$ Pineapple is given as $y_{\text{Pineapple}} = 0$.

(b) The presence of the word Bayesian in the text $X$, that is $S =$ Bayesian, is spuriously correlated with the label 1 (positive sentiment). Therefore, the spurious label associated with $S =$ Bayesian is given as $y_{\text{Bayesian}} = 1$.

The sentiment feature $C$ still remains core within the augmented SS dataset, fully predicting the label $Y$ for each dataset example.

**Causal Graph from this DGP.** The above DGP, through the introduction of spurious feature $S$, induces a causal graph that describes the spurious correlation between spurious feature $S$ and the label $Y$ in terms of the additional variables $X$ and $C$ only. The causal graph, shown in Figure 11b, is similar to the style-content decomposition described in (Kaddour et al., 2022).

**Showing that $\rho_{\text{spurious}}$ Corresponds to the Predictivity of $S$.** We now prove that $\rho_{\text{spurious}}$ gives the co-occurrence rate/predictivity between the label $Y$ and the spurious feature $S$, and is well-defined notation in the sense that it corresponds to Definition 2 so that $\rho_{\text{spurious}} = \mathbb{P}(Y = y_s | S = s)$, where $y_s$ is the label that spurious feature value $s$ is spuriously correlated with. Note that this will hold constant irrespective of the feature value of $S$. We begin with the following proposition.

**Proposition E.1.** *From the DGP described above. we have that*

$$\mathbb{P}(Y = y_s \mid S = s) = \begin{cases} \dfrac{\rho_C \, \rho_{spurious}}{\rho_C \, \rho_{spurious} + (1 - \rho_C)\,(1 - \rho_{spurious})} & \text{if } s = Bayesian, \\[2ex] \dfrac{(1 - \rho_C)\, \rho_{spurious}}{(1 - \rho_C)\, \rho_{spurious} + \rho_C\,(1 - \rho_{spurious})} & \text{if } s = Pineapple. \end{cases} \tag{19}$$

*Proof.* Using the partition theorem, we have that

$$\mathbb{P}(S = s) = \underbrace{\mathbb{P}(S = s \mid \tilde{S} = s)}_{=\mathbb{P}(U_s=1)}\mathbb{P}(\tilde{S} = s) + \underbrace{\mathbb{P}(S = s \mid \tilde{S} \neq s)}_{=\mathbb{P}(U_s=0)}\mathbb{P}(\tilde{S} \neq s) \tag{20}$$

$$= \mathbb{P}(U_s = 1)\mathbb{P}(\tilde{S} = s) + \mathbb{P}(U_s = 0)\mathbb{P}(\tilde{S} \neq s) \tag{21}$$

$$= \rho_{\text{spurious}} \cdot \mathbb{P}(\tilde{S} = s) + (1 - \rho_{\text{spurious}}) \cdot \mathbb{P}(\tilde{S} \neq s) \tag{22}$$

$$= \begin{cases} \rho_C \, \rho_{\text{spurious}} + (1 - \rho_C)\,(1 - \rho_{\text{spurious}}) & \text{if } s = Bayesian, \\ (1 - \rho_C)\, \rho_{\text{spurious}} + \rho_C\,(1 - \rho_{\text{spurious}}) & \text{if } s = Pineapple. \end{cases} \tag{23}$$

where we have used that $\mathbb{P}(U_s = 1) = \rho_{\text{spurious}}$, and that the value of $\tilde{S}$ depends solely on the value of $C$.

Using this alongside Bayes' rule gives

$$\mathbb{P}(Y = y_s \mid S = s) = \frac{\mathbb{P}(S = s \mid Y = y_s)\mathbb{P}(Y = y_s)}{\mathbb{P}(S = s)} \tag{24}$$

$$= \frac{\mathbb{P}(S = s \mid C = y_s)\mathbb{P}(Y = y_s)}{\mathbb{P}(S = s)} \tag{25}$$

$$= \frac{\mathbb{P}(S = s \mid \tilde{S} = s)\mathbb{P}(Y = y_s)}{\mathbb{P}(S = s)} \tag{26}$$

$$= \begin{cases} \dfrac{\rho_C \, \rho_{\text{spurious}}}{\rho_C \, \rho_{\text{spurious}} + (1 - \rho_C)\,(1 - \rho_{\text{spurious}})} & \text{if } s = \text{Bayesian,} \\ \dfrac{(1 - \rho_C)\,\rho_{\text{spurious}}}{(1 - \rho_C)\,\rho_{\text{spurious}} + \rho_C\,(1 - \rho_{\text{spurious}})} & \text{if } s = \text{Pineapple.} \end{cases} \tag{27}$$

which gives the result. $\qquad \square$

**Corollary E.1.** *From the DGP described above, assuming that we have a balanced label distribution, that is $\rho_C = 1/2$, we have that*

$$\mathbb{P}(Y = y_s \mid S = s) = \rho_{\text{spurious}} \, . \tag{28}$$

*for all spurious feature values $s$.*

*Proof.* This follows immediately from he previous proposition, Proposition E.1, with $\rho_C = 1/2$. $\qquad \square$

Within our experiments on the SS datsaet, we always force the label distribution to be balanced, that is $\rho_C = 1/2$, and assume that within each dataset split, $\rho_{\text{spurious}}$ is the same rate for all spurious feature values.

**Data Generation Methodology.** We use Llama-3.1-70B-Instruct to generate modifications $X$ of original dataset examples $\tilde{X}$ to create new text which include the new keyword features (presence of the keywords "Bayesian" and "Pineapple". The prompt we use for generation when modifying examples to include spurious features is give as:

---

**Data Augmentation Prompt**

You are a language model designed to modify a piece of text to include an additional feature in a simple, natural way while keeping your output as similar as possible to the original text.

**Features**
- pineapple: Include the word 'pineapple'.
- Bayesian: Include the word 'Bayesian'.

**Instructions**
- Ensure the output is grammatically correct.
- Keep the output as similar as possible to the original text.
- Make the minimal number of modifications and add the fewest new tokens possible to satisfy the chosen feature.
- Do not change the sentiment of the original text.
- Do not significantly alter the length of the output.
- Incorporate the feature naturally within the original text so that it blends seamlessly with the text's context.
- Do not only append additional clauses at the end of the text to include the feature.
- Inclusions should be case sensitive, e.g., include 'Bayesian' BUT NOT 'bayesian'.

**Output**
- Only return the modified text, with no additional explanations or reasoning.
- Should strictly follow the feature description and the set of instructions.
- Only include the one feature given; the other features SHOULD NOT be included even accidentally.

---

### E.2 Independence Conditions During Training for FIT on SS.

As specified in Appendix D, we would like to have that $Y \perp\!\!\!\perp S$ and $Y_S \perp\!\!\!\perp C$ during training so that models trained via FIT can effectively learn to leverage focus instructions to make predictions based on specified features. Here, $Y_S$ is the spurious label spuriously correlated to the random spurious feature value $S$. The results below give sufficient conditions for these independence conditions to be satisfied with respect to the DGP described above, and consequently form the conditions

that we impose on the SS training set for FIT training. We show that the key condition that we must impose in training for $Y \perp\!\!\!\perp S$ and $Y_S \perp\!\!\!\perp C$ is to set $\rho_{\text{spurious}} = 1/2$.

**Proposition E.2.** *Assuming the DGP described in above and assuming that $\rho_{spurious} = 1/2$, we have that $Y \perp\!\!\!\perp S$.*

*Proof.* With $\rho_{\text{spurious}} = 1/2$, we have from the DGP given above, that $\tilde{S} \perp\!\!\!\perp S$. In particular, using the factorisation implied by the directed acyclic graph (DAG) corresponding to the DGP, we see that

$$\mathbb{P}(Y = y, S = s) = \sum_{c \in \text{Val}(C), \tilde{s} \in \text{Val}(\tilde{S})} \mathbb{P}(Y = y \mid C = c)\mathbb{P}(C = c)\mathbb{P}(\tilde{S} = \tilde{s} \mid C = c)\underbrace{\mathbb{P}(S = s \mid \tilde{S} = \tilde{s})}_{=\mathbb{P}(S=s) \text{ as } S \perp\!\!\!\perp \tilde{S}} \tag{29}$$

$$= \sum_{c \in \text{Val}(C), \tilde{s} \in \text{Val}(\tilde{S})} \mathbb{P}(Y = y \mid C = c)\mathbb{P}(C = c)\mathbb{P}(\tilde{S} = \tilde{s} \mid C = c)\mathbb{P}(S = s) \tag{30}$$

$$= \mathbb{P}(S = s) \sum_{c \in \text{Val}(C), \tilde{s} \in \text{Val}(\tilde{S})} \mathbb{P}(Y = y \mid C = c)\mathbb{P}(C = c)\mathbb{P}(\tilde{S} = \tilde{s} \mid C = c) \tag{31}$$

$$= \mathbb{P}(S = s) \sum_{c \in \text{Val}(C)} \mathbb{P}(Y = y \mid C = c)\mathbb{P}(C = c) \tag{32}$$

$$= \mathbb{P}(S = s)\mathbb{P}(Y = y). \tag{33}$$

By definition, this shows that we have that $Y \perp\!\!\!\perp S$, as required. $\qquad \square$

**Proposition E.3.** *Assuming the DGP described above, for $\rho_{spurious} = 1/2$, then we have that $Y_S \perp\!\!\!\perp C$.*

*Proof.* First note that $Y_S$ is a deterministic function of $S$, that is $Y_S = f(S)$ for some function $f : \text{Val}(S) \to \{0, 1\}$. Therefore, it is sufficient to show that $C \perp\!\!\!\perp S$. However, from the DGP above, we have that $Y = C$. From Proposition E.2, we already have that $Y \perp\!\!\!\perp S$, which implies that $C \perp\!\!\!\perp S$, which proves the claim. $\qquad \square$

### E.3 Results

Figure 12 (a) shows the focus accuracy results of three LLMs on the SS dataset across all of the baseline methods described in Appendix D.4 and the FIT method. We see that across all focus instructions and all models, FIT shows significant improvement over the baselines, achieving very high focus accuracy across all focus instruction types and across all test sets with varying predictivity levels.

*Table 4.* **Complete Baselines vs. FIT Focus Accuracies on SS ($\uparrow$).** For each method, we report the mean of focus-accuracy means ($\mathcal{A}_{\text{focus}}$) over the four test splits ($\mathcal{D}_{\text{iid}}$, $\mathcal{D}_{\text{high}}$, $\mathcal{D}_{\text{low}}$ and $\mathcal{D}_{\text{flipped}}$) $\pm$ the between-split standard deviation of these means (i.e. how performance varies as predictivity changes) across repeats. Boldface indicates the best mean for each focus instruction type and model independently.

| | | $\varnothing$ | focus$(C)$ | focus$(C) \wedge$ ignore$(S)$ | ignore$(S)$ | focus$(S)$ | focus$(S) \wedge$ ignore$(C)$ |
|---|---|---|---|---|---|---|---|
| **Mistral** | Zero-shot | $0.886_{\pm0.007}$ | $0.909_{\pm0.004}$ | $0.899_{\pm0.002}$ | $0.871_{\pm0.006}$ | $0.423_{\pm0.199}$ | $0.305_{\pm0.104}$ |
| | Few-shot | $0.893_{\pm0.003}$ | $0.906_{\pm0.005}$ | $0.904_{\pm0.006}$ | $0.793_{\pm0.036}$ | $0.559_{\pm0.168}$ | $0.634_{\pm0.116}$ |
| | SFT$(y)$ | $0.951_{\pm0.005}$ | $0.950_{\pm0.004}$ | $0.952_{\pm0.005}$ | $0.951_{\pm0.004}$ | $0.445_{\pm0.275}$ | $0.447_{\pm0.270}$ |
| | SFT$(y_{\text{focus}})$ | $0.751_{\pm0.126}$ | $0.794_{\pm0.108}$ | $0.903_{\pm0.042}$ | $0.879_{\pm0.054}$ | $0.506_{\pm0.248}$ | $0.519_{\pm0.241}$ |
| | FIT | $\mathbf{0.953_{\pm0.004}}$ | $\mathbf{0.954_{\pm0.004}}$ | $\mathbf{0.955_{\pm0.004}}$ | $\mathbf{0.954_{\pm0.004}}$ | $\mathbf{0.999_{\pm0.001}}$ | $\mathbf{0.999_{\pm0.001}}$ |
| **Llama** | Zero-shot | $0.500_{\pm0.000}$ | $0.500_{\pm0.000}$ | $0.500_{\pm0.000}$ | $0.500_{\pm0.000}$ | $0.506_{\pm0.003}$ | $0.506_{\pm0.002}$ |
| | Few-shot | $0.674_{\pm0.006}$ | $0.838_{\pm0.008}$ | $0.630_{\pm0.018}$ | $0.508_{\pm0.005}$ | $0.491_{\pm0.047}$ | $0.502_{\pm0.038}$ |
| | SFT$(y)$ | $\mathbf{0.952_{\pm0.004}}$ | $\mathbf{0.954_{\pm0.003}}$ | $0.952_{\pm0.007}$ | $\mathbf{0.952_{\pm0.008}}$ | $0.445_{\pm0.276}$ | $0.444_{\pm0.272}$ |
| | SFT$(y_{\text{focus}})$ | $0.668_{\pm0.178}$ | $0.686_{\pm0.166}$ | $0.803_{\pm0.095}$ | $0.795_{\pm0.094}$ | $0.606_{\pm0.197}$ | $0.601_{\pm0.193}$ |
| | FIT | $0.949_{\pm0.002}$ | $0.951_{\pm0.002}$ | $0.952_{\pm0.002}$ | $0.950_{\pm0.002}$ | $\mathbf{0.998_{\pm0.001}}$ | $\mathbf{0.999_{\pm0.001}}$ |
| **Vicuna** | Zero-shot | $0.420_{\pm0.012}$ | $0.584_{\pm0.007}$ | $0.381_{\pm0.008}$ | $0.355_{\pm0.006}$ | $0.199_{\pm0.105}$ | $0.129_{\pm0.066}$ |
| | Few-shot | $0.147_{\pm0.010}$ | $0.459_{\pm0.014}$ | $0.533_{\pm0.012}$ | $0.431_{\pm0.010}$ | $0.300_{\pm0.150}$ | $0.300_{\pm0.125}$ |
| | SFT$(y)$ | $\mathbf{0.955_{\pm0.005}}$ | $\mathbf{0.956_{\pm0.005}}$ | $0.955_{\pm0.004}$ | $0.953_{\pm0.006}$ | $0.446_{\pm0.278}$ | $0.445_{\pm0.277}$ |
| | SFT$(y_{\text{focus}})$ | $0.570_{\pm0.180}$ | $0.602_{\pm0.187}$ | $0.685_{\pm0.138}$ | $0.671_{\pm0.129}$ | $0.676_{\pm0.129}$ | $0.660_{\pm0.118}$ |
| | FIT | $0.950_{\pm0.003}$ | $0.953_{\pm0.003}$ | $\mathbf{0.955_{\pm0.003}}$ | $\mathbf{0.955_{\pm0.004}}$ | $\mathbf{0.999_{\pm0.001}}$ | $\mathbf{0.999_{\pm0.001}}$ |

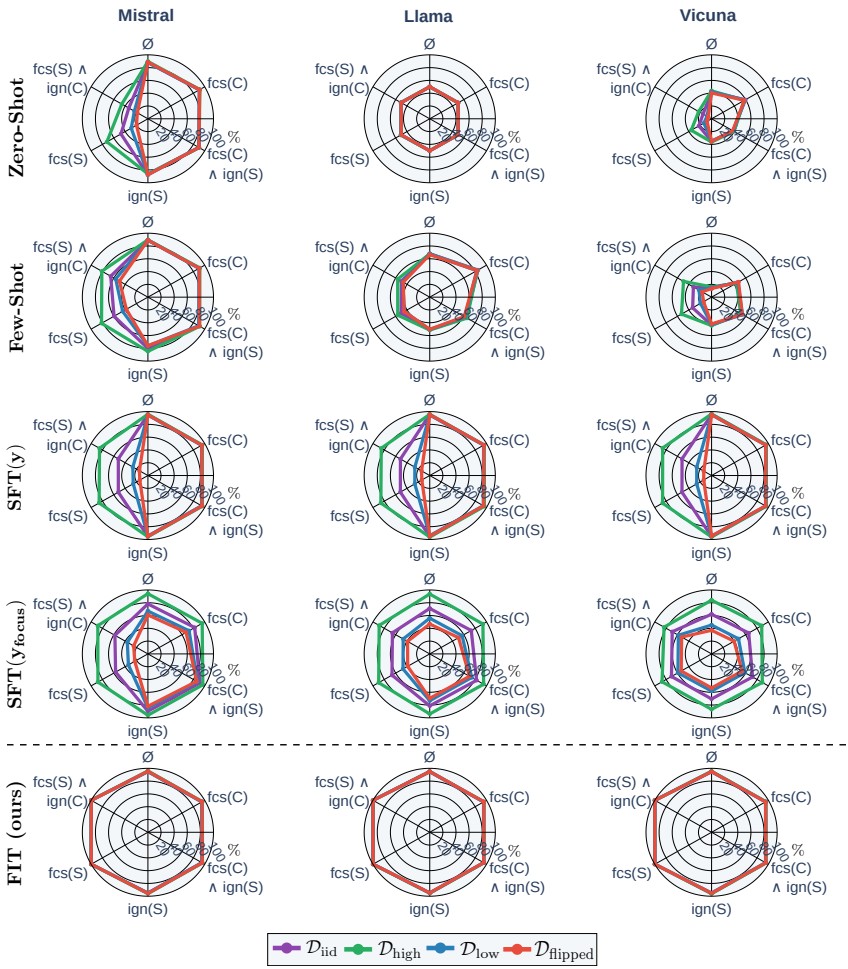

*Figure 12.* **Complete Baseline vs FIT Focus Accuracies on SS (↑).** Figure giving the mean focus accuracies ($\mathcal{A}_{\text{focus}}$) of the additional baselines compared to the focus accuracy of FIT on the SS dataset. The maximum standard deviations of FIT, SFT($y_{\text{focus}}$), SFT($y$) and few-shot methods across models are 2.17%, 14.8%, 0.65%, and 2.83% respectively. fcs = focus, ign = ignore.

> *Key Takeaways.* High focus accuracy on SS indicates that FIT successfully steers model responses based on the feature which it is instructed to focus or to not focus on.

# F Spurious NLI dataset (SMNLI)

We generate a tertiary NLI dataset, SMNLI, with a known spurious feature. We do this subsampling from the original MNLI dataset (Williams et al., 2018). This is a NLI dataset with three labels: entailment (0), neutral (1) and contradiction (2), where data is sampled from 6 underlying categories or genres (government, travel, and fiction, facetoface, nineeleven and verbatim). We aim to induce spurious correlations between the underlying genres and labels.

## F.1 Data-Generating Process (DGP).

We consider a graphical model to describe the DGP of examples within the SMNLI dataset. We use the following variables within our DGP:

- $C$ - NLI relationship between a premise and hypothesis pair, the core feature within this task, sampled from the original MNLI dataset.

- $X$ - example premise and hypothesis from the MNLI dataset.

- $S$ - spurious feature present in the example $X$, here this is the genre of the premise and hypothesis texts. This is a categorical variable.

- $Y$ - final label for example $X$.

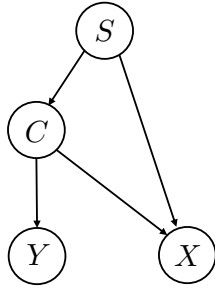

*Figure 13.* **SMNLI DGP**. Graphical model showing the data generating process for subsampling examples from the MNLI dataset to introduce a new spurious keyword feature $S$.

The graphical model described by the DGP for producing the SMNLI dataset is given in Figure 13. Once again, this graphical model can be represented functionally as:

$$S = f_S(U_S); \tag{34}$$
$$C = f_C(S, U_C); \tag{35}$$
$$X = f_X(C, E, U_X); \tag{36}$$
$$Y = f_Y(C, U_Y). \tag{37}$$

More specifically, given the original dataset $\mathcal{D}$ that we are sub-sampling from, the functions that we use within the DGP for the SMNLI dataset are given by:

$$S \sim \mathcal{U}(\mathcal{S}), \tag{38}$$
$$U_C \sim \text{Ber}(\rho_{\text{spurious}}); \tag{39}$$
$$C = U_C \, y_S + (1 - U_C) \, \mathcal{U}(\text{Val}(C) \setminus \{y_S\}) \tag{40}$$
$$X \sim p_{\mathcal{D}}(\cdot | C, S) \tag{41}$$
$$Y = C. \tag{42}$$

Here, $\mathcal{U}(\mathcal{S})$ denotes a uniform distribution over the set of genres $\mathcal{S}$, and we define $\text{Val}(C) = \{0, 1, 2\}$ corresponding to the possible NLI labels for this task. Furthermore, we define $y_S$ to be the NLI label that a particular value of $S$ is spuriously correlated with by design. Moreover, $p_{\mathcal{D}}(\cdot | C, S)$ is the conditional distribution over the dataset examples (premise-hypothesis pairs) that have NLI relationship $C$ and genre $S$.

We restrict the genres that we sample from to $S \in \{\text{government}, \text{fiction}, \text{travel}\}$, a subset of the genres of the training set. When creating a distribution shifted test set, we restrict the genres to $S \in \{\text{facetoface}, \text{nineeleven}, \text{verbatim}\}$. The

specific spurious relationships between genre values $s$ and their associated spurious labels $y_s$ are chosen to be: $y_{\text{slate}} = 0$; $y_{\text{government}} = 2$; $y_{\text{fiction}} = 1$; $y_{\text{travel}} = 0$; $y_{\text{facetoface}} = 2$; $y_{\text{nineeleven}} = 0$; $y_{\text{verbatim}} = 1$. In this way we generate spurious correlations within the dataset through sub-sampling to induce spurious correlations between $S$ and $Y$.

We show that the notion of $\rho_{\text{spurious}}$ in Equation (39) aligns with the notation in Definition 2 and that this does not depend on the spurious feature value of $S$.

**Proposition F.1.** *From the DGP described above, we have that*

$$\mathbb{P}(Y = y_s \mid S = s) = \rho_{spurious} \ . \tag{43}$$

*for all spurious feature values $s$.*

*Proof.* This is clear considering Equation (40), where $\rho_{\text{spurious}}$ influences the chance that we sample $y_s$, i.e. the label spuriously correlated with feature value $s$. $\qquad\square$

### F.2 Independence Conditions During Training for FIT on SMNLI.

As specified in Appendix D, we would like to have that $Y \perp\!\!\!\perp S$ and $Y_S \perp\!\!\!\perp C$ during training so that models trained via FIT can effectively learn to leverage focus instructions to make predictions based on specified features, where, again, $Y_S$ is the label spuriously associated to spurious feature value $S$. The results below give sufficient conditions for this to occur with respect to the DGP described in Figure 13. This boils down to setting $\rho_{\text{spurious}} = 1/3$ during training.

**Proposition F.2.** *Assuming the DGP described above and that $\rho_{spurious} = 1/3$, we have that $Y \perp\!\!\!\perp S$.*

*Proof.* Note that since $Y = C$ in the DGP above, it suffices to show that $C \perp\!\!\!\perp S$. We have that for a given $S = s$, that $\mathbb{P}(C = y_s | S = s) = \rho_{\text{spurious}} = 1/3$ from Proposition F.1. Moreover, let $\tilde{C} \sim \mathcal{U}(\text{Val}(C) \setminus \{y_S\})$ denote the random variable sampled from a uniform distribution over the remaining possible values of $C$ excluding $y_s$. Then for either value of $c \in \text{Val}(C) \setminus \{y_s\}$ we have that

$$\mathbb{P}(C = c | S = s) = \mathbb{P}(U_C = 0)\mathbb{P}(\tilde{C} = c) \tag{44}$$

$$= \frac{2}{3} \cdot \frac{1}{2} \tag{45}$$

$$= \frac{1}{3}. \tag{46}$$

Therefore, we have that $C \sim \mathcal{U}(\text{Val}(C))$. In particular, this then gives that $C \perp\!\!\!\perp S$, which in turn implies that $Y \perp\!\!\!\perp S$. $\quad\square$

**Proposition F.3.** *Assuming the DGP described above and that $\rho_{spurious} = 1/3$, then we have that $Y_S \perp\!\!\!\perp C$.*

*Proof.* Note that $Y_S$ is a deterministic function of $S$, so that $Y_S = f(S)$ for some function $f : Val(S) \rightarrow \text{Val(Y)}$. Therefore, it suffices to show that $S \perp\!\!\!\perp C$. The proof of this is given in Proposition F.2. $\qquad\square$

### F.3 Results.

We present full results comparing FIT on SMNLI against all of the baselines described in Appendix D.4 in Figure 14, Table 5 for completeness.

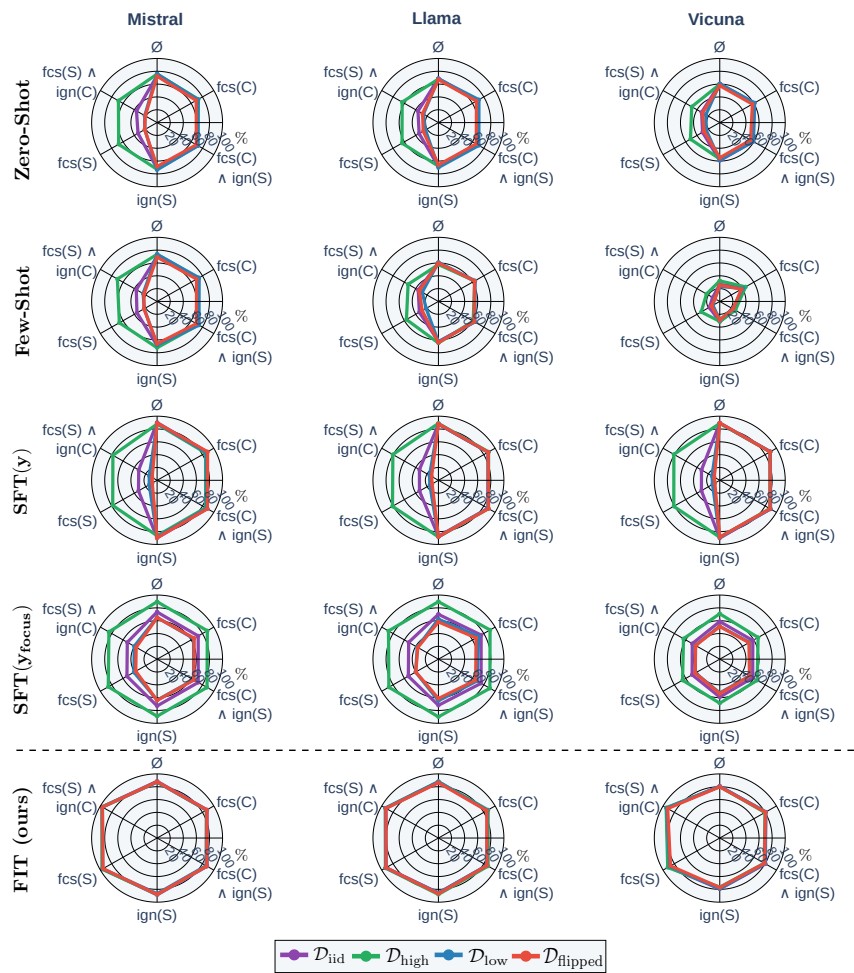

*Figure 14.* **Complete Baseline vs FIT Focus Accuracies on SMNLI (↑).** Figure giving focus accuracies ($\mathcal{A}_{\text{focus}}$) of the additional baselines compared to the focus accuracy of FIT on the SMNLI dataset. The maximum standard deviations of FIT, SFT($y_{\text{focus}}$), SFT($y$) and few-shot methods across models are 6.47%, 7.98%, 1.15%, and 0.500% respectively. fcs = focus, ign = ignore.

*Table 5.* **Complete Baselines vs. FIT Focus Accuracies on SMNLI (↑).** For each method, we report the mean of focus-accuracy means ($\mathcal{A}_{\text{focus}}$) over the four test splits ($\mathcal{D}_{\text{iid}}$, $\mathcal{D}_{\text{high}}$, $\mathcal{D}_{\text{low}}$ and $\mathcal{D}_{\text{flipped}}$) $\pm$ the between-split standard deviation of these means (i.e. how performance varies as predictivity changes) across repeats. Boldface indicates the best mean for each focus instruction type and model independently.

| | | $\varnothing$ | focus($C$) | focus($C$) $\wedge$ ignore($S$) | ignore($S$) | focus($S$) | focus($S$) $\wedge$ ignore($C$) |
|---|---|---|---|---|---|---|---|
| **Mistral** | Zero-shot | $0.742_{\pm 0.016}$ | $0.711_{\pm 0.014}$ | $0.711_{\pm 0.012}$ | $0.716_{\pm 0.020}$ | $0.362_{\pm 0.189}$ | $0.368_{\pm 0.191}$ |
| | Few-shot | $0.716_{\pm 0.020}$ | $0.725_{\pm 0.017}$ | $0.722_{\pm 0.017}$ | $0.687_{\pm 0.023}$ | $0.370_{\pm 0.178}$ | $0.386_{\pm 0.189}$ |
| | SFT($y$) | $\mathbf{0.890}_{\pm 0.009}$ | $0.875_{\pm 0.012}$ | $\mathbf{0.875}_{\pm 0.013}$ | $\mathbf{0.885}_{\pm 0.013}$ | $0.334_{\pm 0.275}$ | $0.332_{\pm 0.273}$ |
| | SFT($y_{\text{focus}}$) | $0.732_{\pm 0.101}$ | $0.727_{\pm 0.097}$ | $0.725_{\pm 0.096}$ | $0.726_{\pm 0.102}$ | $0.544_{\pm 0.192}$ | $0.537_{\pm 0.190}$ |
| | FIT | $0.878_{\pm 0.005}$ | $\mathbf{0.875}_{\pm 0.004}$ | $0.874_{\pm 0.006}$ | $0.878_{\pm 0.007}$ | $\mathbf{0.963}_{\pm 0.006}$ | $\mathbf{0.968}_{\pm 0.003}$ |
| **Llama** | Zero-shot | $0.679_{\pm 0.007}$ | $0.688_{\pm 0.017}$ | $0.682_{\pm 0.018}$ | $0.682_{\pm 0.015}$ | $0.370_{\pm 0.162}$ | $0.386_{\pm 0.151}$ |
| | Few-shot | $0.595_{\pm 0.012}$ | $0.641_{\pm 0.003}$ | $0.630_{\pm 0.010}$ | $0.636_{\pm 0.009}$ | $0.351_{\pm 0.130}$ | $0.379_{\pm 0.094}$ |
| | SFT($y$) | $\mathbf{0.880}_{\pm 0.005}$ | $\mathbf{0.880}_{\pm 0.003}$ | $\mathbf{0.879}_{\pm 0.004}$ | $\mathbf{0.879}_{\pm 0.004}$ | $0.352_{\pm 0.278}$ | $0.354_{\pm 0.272}$ |
| | SFT($y_{\text{focus}}$) | $0.700_{\pm 0.121}$ | $0.759_{\pm 0.093}$ | $0.750_{\pm 0.098}$ | $0.720_{\pm 0.110}$ | $0.552_{\pm 0.194}$ | $0.536_{\pm 0.209}$ |
| | FIT | $0.872_{\pm 0.007}$ | $0.865_{\pm 0.008}$ | $0.861_{\pm 0.009}$ | $0.865_{\pm 0.010}$ | $\mathbf{0.929}_{\pm 0.005}$ | $\mathbf{0.931}_{\pm 0.004}$ |
| **Vicuna** | Zero-shot | $0.595_{\pm 0.012}$ | $0.598_{\pm 0.015}$ | $0.575_{\pm 0.019}$ | $0.573_{\pm 0.017}$ | $0.334_{\pm 0.108}$ | $0.341_{\pm 0.093}$ |
| | Few-shot | $0.269_{\pm 0.032}$ | $0.417_{\pm 0.027}$ | $0.251_{\pm 0.019}$ | $0.301_{\pm 0.013}$ | $0.195_{\pm 0.080}$ | $0.133_{\pm 0.060}$ |
| | SFT($y$) | $\mathbf{0.892}_{\pm 0.003}$ | $\mathbf{0.890}_{\pm 0.002}$ | $\mathbf{0.890}_{\pm 0.004}$ | $\mathbf{0.891}_{\pm 0.006}$ | $0.337_{\pm 0.288}$ | $0.342_{\pm 0.283}$ |
| | SFT($y_{\text{focus}}$) | $0.579_{\pm 0.081}$ | $0.575_{\pm 0.064}$ | $0.573_{\pm 0.055}$ | $0.585_{\pm 0.063}$ | $0.502_{\pm 0.092}$ | $0.496_{\pm 0.086}$ |
| | FIT | $0.803_{\pm 0.004}$ | $0.807_{\pm 0.006}$ | $0.794_{\pm 0.008}$ | $0.781_{\pm 0.012}$ | $\mathbf{0.894}_{\pm 0.019}$ | $\mathbf{0.934}_{\pm 0.010}$ |

# G   Additional BBQ Results

In Figure 15 and Table 6 we include the full comparisons of FIT against all baselines described in Appendix D.4 on BBQ.

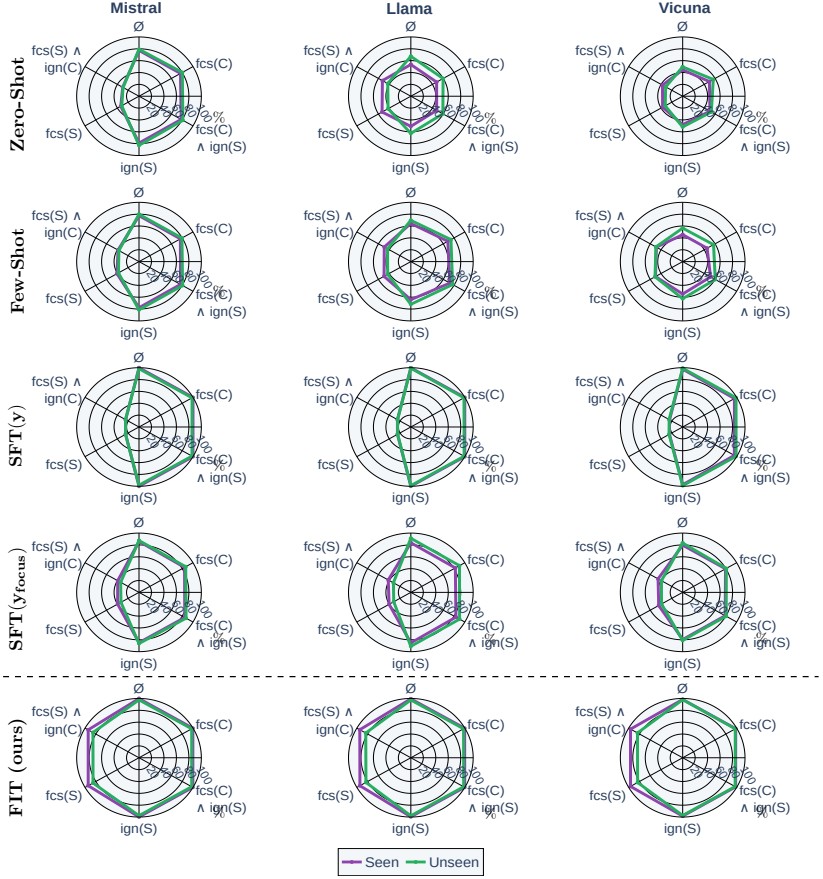

*Figure 15.* **Complete Baseline vs FIT Focus Accuracies on BBQ** (↑)**.** Figure giving focus accuracies ($\mathcal{A}_{\text{focus}}$) of the additional baselines compared to the focus accuracy of FIT on the BBQ dataset. The maximum standard deviations of FIT, SFT($y_{\text{focus}}$), SFT($y$) and few-shot methods across models are 4.07%, 10.7%, 2.22%, and 0.600% respectively. fcs = focus, ign = ignore.

*Table 6.* **Complete Baseline vs FIT Focus Accuracies on BBQ** (↑)**.** We report across all baselines the mean seen/unseen focus accuracies, with boldface indicating the best mean for each focus instruction type and model independently. The maximum standard deviations of FIT, SFT($y_{\text{focus}}$), SFT($y$) and few-shot methods across models are 4.07%, 10.7%, 2.22%, and 0.600% respectively. fcs = focus, ign = ignore.

| | | $\varnothing$ | focus($C$) | focus($C$) ∧ ignore($S$) | ignore($S$) | focus($S$) | focus($S$) ∧ ignore($C$) |
|---|---|---|---|---|---|---|---|
| **Mistral** | Zero-shot | 0.772/0.792 | 0.768/0.803 | 0.782/0.812 | 0.790/0.829 | 0.338/0.320 | 0.302/0.295 |
| | Few-shot | 0.775/0.798 | 0.767/0.795 | 0.769/0.807 | 0.777/0.814 | 0.408/0.381 | 0.388/0.373 |
| | SFT($y$) | **0.999/0.982** | **1.000/0.983** | **1.000/0.982** | **1.000/0.981** | 0.248/0.249 | 0.248/0.249 |
| | SFT($y_{\text{focus}}$) | 0.858/0.874 | 0.844/0.866 | 0.848/0.872 | 0.847/0.863 | 0.396/0.335 | 0.394/0.340 |
| | FIT | 0.998/0.976 | 0.998/0.976 | 0.998/0.977 | 0.999/0.976 | **0.941/0.852** | **0.941/0.853** |
| **Llama** | Zero-shot | 0.534/0.673 | 0.478/0.597 | 0.478/0.593 | 0.508/0.627 | 0.533/0.421 | 0.527/0.428 |
| | Few-shot | 0.643/0.696 | 0.686/0.743 | 0.724/0.774 | 0.642/0.720 | 0.496/0.429 | 0.492/0.432 |
| | SFT($y$) | **0.998/0.991** | **0.999/0.985** | **1.000/0.988** | **0.999/0.989** | 0.248/0.244 | 0.248/0.246 |
| | SFT($y_{\text{focus}}$) | 0.836/0.912 | 0.825/0.901 | 0.836/0.903 | 0.852/0.910 | 0.403/0.317 | 0.418/0.323 |
| | FIT | 0.993/0.974 | 0.997/0.977 | 0.998/0.979 | 0.996/0.979 | **0.943/0.832** | **0.946/0.835** |
| **Vicuna** | Zero-shot | 0.444/0.495 | 0.495/0.567 | 0.505/0.531 | 0.479/0.519 | 0.371/0.324 | 0.375/0.315 |
| | Few-shot | 0.454/0.565 | 0.453/0.572 | 0.517/0.604 | 0.547/0.625 | 0.503/0.512 | 0.487/0.496 |
| | SFT($y$) | 0.975/**0.991** | 0.959/**0.988** | 0.957/**0.990** | 0.974/**0.990** | 0.245/0.252 | 0.243/0.251 |
| | SFT($y_{\text{focus}}$) | 0.796/0.832 | 0.794/0.808 | 0.787/0.806 | 0.796/0.809 | 0.444/0.392 | 0.456/0.394 |
| | FIT | **0.983**/0.979 | **0.982**/0.975 | **0.981**/0.972 | **0.984**/0.973 | **0.966/0.835** | **0.966/0.837** |

# H    BBQ-NLG

## H.1    BBQ-NLG Experimental Setup

We follow the original BBQ dataset splits (including both training and held-out social-bias partitions) described in Section 4.2. For each example in the new BBQ-NLG dataset, we remove the predefined answer choices and require the model to generate fully verbalised responses, rather than selecting from a fixed set of three options as in the original BBQ dataset and outputting a categorical label.

To give the model additional capacity for this more challenging free-form generation task, we augment the usual LoRA targets (the key and value projection matrices) with adapters on the value-projection matrices as well. All LoRA modules are trained with a learning rate of $10^{-5}$. We train both the FIT and SFT($y_{\text{focus}}$) models for 10 epochs each. As a strong comparison, we evaluate a few-shot baseline that conditions the model on 4 in-context examples that share the same focus instruction as the test instance.

To enable both rapid and cost-effective evaluation of correctness in our open-ended generation task, we employ an LLM-based judge. Specifically, we use a pre-trained Llama-3.1-8B-Instruct model to compare each generated response against its associated ground-truth focus label and determine whether they are semantically equivalent. Manual checks confirm that the Llama-3.1-8B-Instruct judge reliably assesses semantic equivalence in this setting where model generations and expected responses and generally short and concise.

## H.2    Results

Figure 16 reports the focus-accuracy of each method across models. Although all approaches exhibit a slight degradation on unseen feature combinations relative to the classification-style BBQ results in Section 4.2, FIT remains generally overall on par with the earlier numbers. Critically, FIT consistently outperforms both the SFT($y_{\text{focus}}$) and the strong 4-shot few-shot baseline that uses identical focus prompts at test time. These findings provide clear evidence that the FIT method can be effectively extended to open-ended NLG settings without sacrificing its steering capabilities.

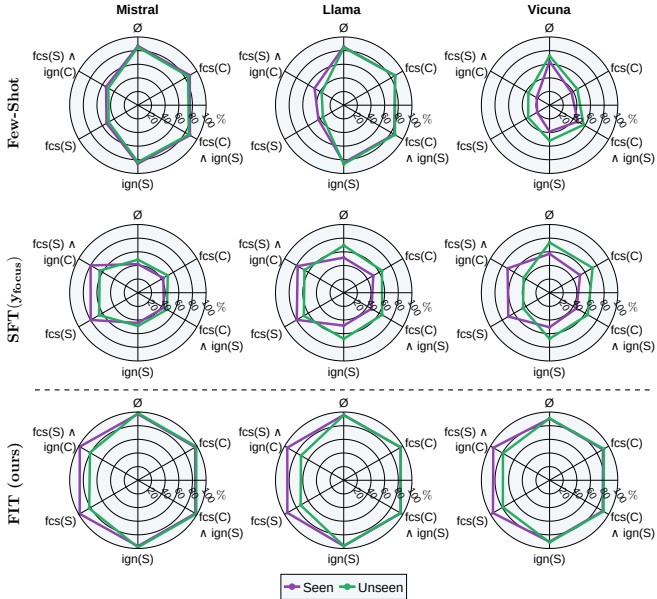

*Figure 16.* **BBQ-NLG Focus Accuracies** ($\uparrow$). Mean focus accuracy ($\mathcal{A}_{\text{focus}}$) of baselines and FIT on the BBQ-NLG dataset. The maximum standard deviations of FIT, SFT($y_{\text{focus}}$) and few-shot methods across models and $\mathcal{I}_{\text{focus}}$ are 2.45%, 6.35% and 0.377% respectively. fcs = focus, ign = ignore.

# I  Comparison of FIT Against a Specific Debiasing Technique

FIT is a general framework designed to enable users to steer a model's behaviour based on specified features. This approach provides enhanced control over model outputs during inference, adding a critical layer of explainability and controllability to model predictions.

While understanding and mitigating biases or spurious correlations is a valuable and natural application of FIT, it is not the sole objective. The broader goal of steerability includes addressing challenges in managing and aligning model behaviour across diverse contexts. For instance, maintaining controllability is crucial in addressing safety alignment fragility, which can emerge after fine-tuning (Bhattacharjee et al., 2024). In such cases, the ability to adapt model responses to align with user specifications ensures safe and reliable deployment.

**Experiment.** To explore FIT's broader applicability, we compare its performance as a debiasing method against a well-known debiasing technique: the Product of Experts (PoE) method (Mahabadi et al., 2020). PoE involves training a bias model $f_B$, which is trained exclusively on bias features. This bias model mediates the training of the final model $f$ by combining their predictions through an elementwise product: $\sigma(f(x)) \odot \sigma(f_B(x_B))$, where $x \in \mathcal{D}$, for dataset $\mathcal{D}$, and $x_B$ represents the biased feature of $x$.

We adapted this approach to our setting by training a bias model on the stereotypical labels within the BBQ dataset. These labels correspond to group-stereotypical associations. For autoregressive models, we further modified the PoE method by extracting and normalising the logits of the first newly generated token position over the set of single tokens representing the answer options.

**Results.** The results of the debiasing experiment comparing FIT to the PoE method is shown in Figure 17. FIT performs equally as well as the PoE method as shown by comparing the default prompt accuracy ($\emptyset$) for the PoE models against the focus($C$) results for the FIT models; both metrics correspond to causal accuracy for these prompt types. This indicates that FIT performs just as well as the PoE dedicated debiasing technique.

However, the PoE method requires training two separate models and does not provide steerability at test time as shown by the low focus accuracy on focus($S$). Indeed the model defaults to the ground truth label across all prompt types and does not change behaviour in the presence of different focus specifications. This highlights the flexibility of FIT, which not only debiases effectively but also enables additional controllability during inference.

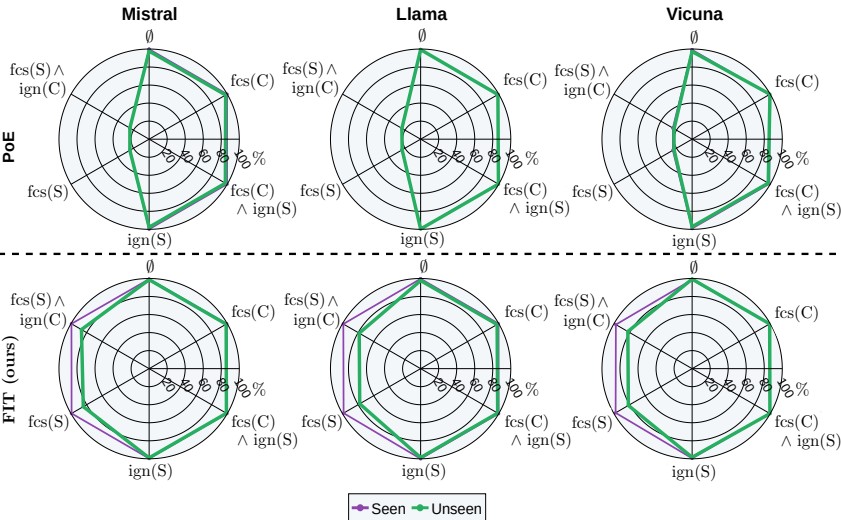

*Figure 17.* **Focus Accuracy of FIT Against PoE Debiasing Technique (↑).** Figure showing the focus accuracies ($\mathcal{A}_{\text{focus}}$) of FIT (bottom row) and the dedicated debiasing technique, PoE (top row), on the BBQ dataset.

# J   Spurious HANS Dataset (SHANS)

We generate a binary NLI dataset, SHANS, with a known spurious feature. We do this by subsampling from the HANS dataset (McCoy et al., 2019). This is an NLI data set with two labels: entailment (0) and contradiction (1). This is an adversarial dataset designed to assess different NLI models' reliance on spurious heuristics rather than on the underlying relationship between the premise and the hypothesis when making NLI predictions. Specifically, the author's consider three major categories of heuristics: lexical overlap heuristic (assuming that a premise entails from words within the hypothesis) , sub-sequence heuristic (assuming that the premise entails all any of its contiguous sub-sequences of words) and constituent heuristic (assuming that a premise entails a hypothesis that is any constituent within it's syntactic parse tree). We take each of these as separate spurious features within our SHANS dataset for which we induce spurious correlations, as for the SMNLI dataset, through subsampling.

## J.1   Data-Generating Process (DGP).

We consider a graphical model to describe the DGP of examples within the SHANS dataset. We use the following variables within our DGP:

- $C$ - NLI relationship between a premise and hypothesis pair, the core feature within this task, sampled from the original HANS dataset.

- $S_{\text{lex.}}$ - spurious feature, here the presence of a hypothesis entirely made from words from the premise. This is a binary categorical variable (present/not present).

- $S_{\text{sub.}}$ - spurious feature, here the presence of a hypothesis that is a contiguous subsequence of the premise. This is a binary category feature (present/not present).

- $S_{\text{const.}}$ - spurious feature, here the presence of hypothesis that is a constituent/subtree of the premise. Here we have a binary variable (present/not present).

- $X$ - example from the HANS dataset.

- $Y$ - final label for example $X$.

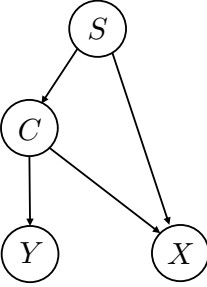

*Figure 18.* **SHANS DGP**. Graphical model showing the DGP for subsampling examples from the SHANS dataset to introduce new spurious features $S_{\text{lex.}}$, $S_{\text{sub.}}$ and $S_{\text{const.}}$ which are encoded within the categorical spurious feature $S$ which represents one of these three heuristics.

The graphical model described by the DGP for producing the S-HANS dataset is given in Figure 18. Once again, this graphical model can be represented functionally as

$$S = f_S(U_S); \tag{47}$$
$$C = f_C(S, U_C); \tag{48}$$
$$X = f_X(C, E, U_X); \tag{49}$$
$$Y = f_Y(C, U_Y), \tag{50}$$

where here we define $S$ to be a categorical feature over the set of variables indicating the presence of each of the three heuristics introduced above which we denote, through overloaded notation, by $\mathcal{S} = \{s_{\text{lex.}}, s_{\text{sub.}}, s_{\text{const.}}\}$. More specifically,

given the original HANS dataset $\mathcal{D}$ that we are sub-sampling from, the functions that we use within the DGP for the SHANS dataset are given by:

$$S \sim \mathcal{U}(\mathcal{S}), \tag{51}$$

$$U_C \sim \text{Ber}(\rho_{\text{spurious}}); \tag{52}$$

$$C \sim U_C y_S + (1 - U_C)(1 - y_S); \tag{53}$$

$$X \sim p_{\mathcal{D}}(\cdot | C, S) \tag{54}$$

$$Y = C. \tag{55}$$

Here, $\mathcal{U}(\mathcal{S})$ is a uniform categorical distribution over the set of spurious features $\mathcal{S}$ which effectively selects the presence of exactly one of the three spurious feature heuristics. We define $y_S$ to be the NLI label that a particular value of $S$ is spuriously correlated with by design. Moreover, $p_{\mathcal{D}}(\cdot | C, S)$ is the conditional distribution over the dataset examples (premise-hypothesis pairs) that have NLI relationship $C$ and the presence of spurious heuristic $S$.

We restrict the genres to $S \in \{s_{\text{lex.}}, s_{\text{const.}}\}$ for heuristics seen during training. We then create additional test sets with spurious feature set restricted to $S \in \{s_{\text{sub.}}\}$, which serves as an unseen spurious feature set to test generalisation.

We consider the presence of each feature to be separate binary spurious features. The specific spurious correlations between heuristics and labels $Y$ are chosen to be: $y_{S_{\text{lex.}}=1} = 0$; $y_{S_{\text{sub.}}=1} = 0$; $y_{S_{\text{const.}}=1} = 1$. In this way we generate spurious correlations within the dataset through sub-sampling to induce spurious correlations between the heuristics $S$ and label $Y$.

### J.2 Transferred Results from SMNLI.

As we have effectively used the same DGP as for the SMNLI dataset described in Appendix F, with the only change being the label set, all of the results that we have proven for SMNLI, translate to the SHANS dataset. In particular, we have that $\rho_{\text{spurious}}$ aligns with the notation in Definition 2, and that we have $Y \perp\!\!\!\perp S$ and $Y_S \perp\!\!\!\perp C$ under the assumptions of $\rho_{\text{spurious}} = 1/2$ within the training set.

# K FIT on SHANS

Here, we give the results performing FIT on the SHANS dataset.

## K.1 Spurious HANS (SHANS) Dataset.

We generate binary NLI dataset sub-sampled from HANS (McCoy et al., 2019), a dataset designed to challenge NLI models by exposing common heuristics they rely on, such as lexical overlap (whether the hypothesis shares many words with the premise), sub-sequence (whether the hypothesis is a contiguous sub-sequence of the premise), and constituent (whether the hypothesis is a grammatical sub-structure of the premise). The presence of these heuristics are spuriously correlated with labels through sub-sampling of the presence of each of the heuristics from the original dataset. The degree of co-occurrence is governed by $\rho_{\text{spurious}}$, which varies according to the test sets described in Section 3. We ensure that $\rho_{\text{spurious}}$ is the same for all feature values within each dataset split. In particular, we set $\rho_{\text{spurious}}$ to be $0.5, 0.5, 0.9, 0.25$ and $0.9$ (in this case with flipped spurious correlations) on $\mathcal{D}_{\text{train}}, \mathcal{D}_{\text{iid}}, \mathcal{D}_{\text{high}}, \mathcal{D}_{\text{low}}$ and $\mathcal{D}_{\text{flipped}}$ respectively. We keep the subsequence heuristic as a held-out unseen feature during training for testing FIT generalisation.

## K.2 Results.

Figure 19 shows the focus accuracy results of performing FIT on the SHANS dataset for the Llama-3.1-8B-Instruct model. As expected, on the seen features, FIT shows high focus accuracy across all focus instructions. However, for unseen features, we observe lower focus accuracy. This could be attributed to the overlapping nature of the heuristics in SHANS, which are often graded versions of each other with different levels of specificity. For instance, the sub-sequence heuristic can overlap with both lexical overlap and constituent heuristics (e.g., the example with Premise: "Before the actor slept, the senator ran" and Hypothesis: "The actor slept." satisfies all three heuristics). This overlap likely confuses the model during generalisation, as it struggles to distinguish between heuristics seen during training and those that were not. These results suggest a potential limitation of FIT when dealing with features that are not sufficiently distinct or have significant overlap.

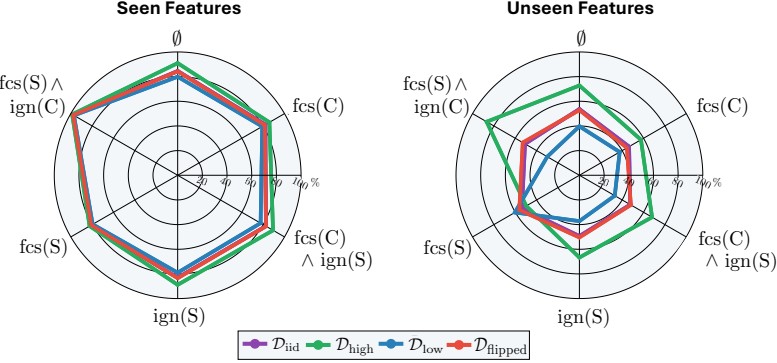

*Figure 19.* **SHANS Focus Accuracies** ($\uparrow$). Focus accuracy ($\mathcal{A}_{\text{focus}}$) of Llama-3.1-8B-Instruct after FIT on the SHANS test datasets containing either the seen or unseen spurious features during training. Here, $C$ refers to the core feature (logical relationship between premise and hypothesis) and $S$ the spurious feature (heuristic used).

