# OpenReview forum: "Focus On This, Not That! Steering LLMs with Adaptive Feature Specification"
_ICML.cc/2025/Conference — ICML 2025 poster_

### Official Review · Reviewer_2pfW · 2025-03-13

**Overall Recommendation:** 4

**Summary:**

The paper proposes a modification to the typical instruction-tuning process by including a "focus prompt" in the context, to guide the model to focus on certain aspects of the user input. Experiments on two synthetic settings demonstrated the effectiveness of the proposed method over vanilla SFT and few-shot baselines.

**Claims And Evidence:**

It is evident that including a focus prompt in the instruction will help with focusing on certain parts of the user input. It is unclear why training (instruction-tuning) on such data is necessary to achieve said effect. See more elaboration in the "Questions For Authors" section below.

**Essential References Not Discussed:**

Not to the extent of my knowledge.

**Experimental Designs Or Analyses:**

Yes, but there are important baselines missing, specifically vanilla SFT + testing focus instruction prompting and directly rewriting the user input with the spurious features removed. Another missing baseline is the typical conditional supervised fine-tuning, by appending control tokens in place of the focusing prompt.

**Methods And Evaluation Criteria:**

The toy experiment on SS with synthetic keyword labels of {Bayesian, Pineapple} is a bit too artificial. The choice of keyword is so specific in a way that it almost seems like the keywords are cherry-picked. Since the experiment shouldn't be expensive, consider expanding the datasets to validate the method on with the additional baselines mentioned in the "Experimental Designs Or Analyses" section below.

**Other Comments Or Suggestions:**

N/A

**Other Strengths And Weaknesses:**

N/A

**Questions For Authors:**

My main concern of this work is the relevance of the technique. A major assumption that FIT makes is knowing which features are spurious beforehand (even before the training phase). If we do have the information of which features are spurious, there are a number of easier techniques we could apply to prevent the spurious prediction. One is prompting the model during testing with focus instructions. A well-trained instruction-tuned should be able to properly ignore certain parts of the input to make the prediction. Another even more fundamental solution is to first have the language model rewrite the question while leaving out the spurious piece of information, then respond to the augmented question. These are all fairly straightforward techniques that don't involve complicating the training process. So unless the authors can justify why modifying the instruction tuning process to include the focus prompt is absolutely critical, it seems to be an unnecessary complication of the problem.

One other concern is the scalability of FIT with respect to multiple potentially spurious features. In the experiments, the setting involves one very specific set of spurious features (e.g. {Bayesian, Pineapple}). In non-synthetic scenarios, there are multiple potential spurious features we would hope to control independently during inference time. It would be devastating if FIT requires retraining the model every time a new spurious feature is introduced. It would also be infeasible to maintain multiple copies of the model for different spurious relation protections. Thus, it is important that the method scales efficiently with the number of spurious features and can effectively switch between different combinations of the features. This part is lacking in the paper.

Another concern is the sensitivity to the focus instruction during test time. During test time, sometimes we don't wish the model to focus of certain features. In this case, would the FITed model be over-reliant on the focus prompt during inference time to make the correct prediction. If the focus instruction is removed, would the model still perform similarly to the vanilla SFTed version? Are there tradeoffs to the performance with the model being over-sensitive to the focus instruction?

My final concern is regarding the assumption of oracle access to knowing which features are spurious beforehand. This is a very strong assumption and consequently a lot of simpler techniques could be applied if we do have such information (see additional baselines in the first paragraph). Most cases we don't have the information of the spurious features. One could argue that this is out of scope of this paper and that might be true, but some heuristics to identify spurious features should also be provided to make FIT generally useful.

**Relation To Broader Scientific Literature:**

Typically people solve the issue of spurious features when querying LLMs with prompting approaches. Fine-tuning on focusing prompts have not been explicitly investigated. However, FIT can be viewed as a specialized version of conditional supervised fine-tuning (cSFT) where control tokens are appended to steer the model to behave in certain ways.

**Theoretical Claims:**

N/A

---

> ### Author Rebuttal · Authors · 2025-03-31
>
> ## P1: Spurious Feature Knowledge and Baselines
>
> **Simpler Baselines suggested by the reviewer:**
>
> - **Test-Time Prompting using focus instructions without FIT training:** Already tested in our paper via:
>   - **Zero-Shot** (lines 318–321, Figures 6–8)
>   - **SFT(y)** (line 1053, Figures 6–8)
>
>   *Result:* These methods fail to reliably steer model outputs without FIT training, indicating that models do indeed struggle to ignore certain parts of the input, showing that the inclusion of focus instruction for FIT is necessary.
>
> - **Input Rewriting:** Impractical for complex datasets (e.g., SMNLI), as spurious features (like genre) are usually latent, and deeply embedded and cannot be easily rewritten without altering the underlying example and could lead to label mismatch in the process. FIT is therefore a lot more practical and simpler in these more complex and realistic settings.
>
> **Prior Knowledge of Spurious Features:**
> Identifying spurious features is not a practical limitation of FIT as it  aligns with standard industry practices for transparent, reliable modeling, and moreover existing methods such as automated spurious correlation detectors can be used alongside FIT (see our detailed discussion in Appendix B, which also includes heuristics for identifying spurious features, as requested).
>
>
> ## P2: Scalability and Retraining
>
> **Scalability:** FIT does not require retraining for every new spurious feature. Experiments on BBQ (Section 4.2) and in NLG settings (see response P1 to reviewer ivne)  confirm FIT’s ability to adapt at test time to unseen features without retraining or providing additional knowledge.
>
> **Multiple Features:** Our current experiments provide initial evidence that FIT handles combined instructions (i.e., the presence of both 1 focus and 1 ignore type in the same specification), and we will expand this in future work. We agree that focusing on and/or ignoring arbitrarily many features simultaneously is an important direction for future work, and will further highlight this in the future work section.
>
>
> ## P3: Sensitivity to Focus Instructions
>
> Our paper already addresses this concern:
>
> - **Default Prompt Performance:** FIT maintains comparable accuracy to SFT models even without explicit instructions (Figures 6–8) - compare default/empty focus types and does not show sensitivity when dropping focus instructions (compare default/ empty to focus(C) focus types, corresponding to task casual accuracy). Moreover, the ablation in line [391, right] shows that FIT does not harm pre-existing instruction-following capabilities.
> - **Robustness to Wording Variations:** Ablation studies (Section 5 line 430, left) show variations in instruction wording between train- and test-time prompts have minimal impact.
>
>
> ## P4: Comparison with Conditional SFT (cSFT) and Control Token Variants
>
> We appreciate the reviewer suggesting additional related work, though specific references were not provided. We identified two potentially relevant methods: conditional SFT (cSFT) ([Zhang et al., 2024](https://arxiv.org/abs/2406.01976)) and the control token variant SteerLM ([Dong et al., 2023](https://arxiv.org/abs/2310.05344)). FIT significantly differs from these approaches as follows:
>
> **Conditional SFT (cSFT):**
> - **Objective:** cSFT prevents corpus-level spurious correlations but lacks dynamic adaptability at test-time.
> - **FIT Difference:** FIT explicitly trains for flexible test-time adaptability via natural language instructions, dynamically generalising to unseen features without retraining (see Section 4.3 BBQ experiments).
>
> **Control Token Methods (e.g., SteerLM):**
> - **Data Annotation:** SteerLM uses fixed, human-annotated stylistic attributes (e.g., humor), focusing on output style. FIT directly annotates inputs with dynamic instructions to prioritize input features.
> - **Feature Specification:** SteerLM uses predefined attribute values, limiting flexibility. FIT dynamically detects and prioritizes input features through natural language prompts, without explicit attribute values.
> - **Training:** SteerLM relies on iterative bootstrapping. FIT employs straightforward supervised fine-tuning without bootstrapping.
> - **Prompting Mechanism:** SteerLM's fixed attribute tokens limit its adaptability. FIT uses flexible natural language prompts enabling real-time steering and generalisation to unseen features at test-time.
> - **Goals:** SteerLM aims for stylistic output adjustments, whereas FIT directly enhances model robustness, fairness, and alignment by controlling input-feature relevance.
>
> **Summary:** FIT fundamentally differs from cSFT and control-token variants. These distinctions will be clearly detailed in the final paper. Please let us know if there is a specific cSFT paper you had in mind, and we are happy to further detail its relationship to FIT.
>
> ## Conclusion and Final Comments
>
> We appreciate the reviewer’s valuable feedback. We hope our clarifications fully address the concerns raised.

---

> > ### Comment · Reviewer_2pfW · 2025-04-04
> >
> > Dear authors,
> >
> > Thank you for the clarification. Many points raised in the review were due to misreading the paper and I take full responsibility. The radar diagrams are slightly harder to parse than, say tables, but ultimately they do convey the supporting evidence for the claims made in the paper.
> >
> > One crucial future work direction is extending to tasks beyond text classification. For example, text-conditioned image generation should benefit significantly from similar techniques for control that is not restricted by the data distribution.

---

> > > ### Author Response · Authors · 2025-04-04
> > >
> > > Dear Reviewer,
> > >
> > > Thank you for your updated feedback.
> > >
> > >  - We agree the radar plots alone are not the easiest to parse, so we will include full tables in the appendix of the final version of the paper and point to these from the main paper for clarity.
> > >
> > > - Finally, we agree the extension of FIT to text-conditioned image generation is definitely and interesting future direction. We will include a mention of this within our future work section.
> > >
> > > Thank you again for your helpful comments and responsiveness during the reviewer process.

---

### Official Review · Reviewer_4nt1 · 2025-03-15

**Overall Recommendation:** 4

**Summary:**

The main aim of this work is to finetune language models to be robust to known spurious correlations and features.  To this end, Focus Instruction Tuning (FIT) is introduced, a method for instruction tuning language models with prioritization towards certain features, and “ignoring” others.  the approach is based on adding two sets of focus and ignore labels to the prompt for the model.  The method is supported by some theoretical analysis and experiments on classification tasks such as NLI and QA.

**Claims And Evidence:**

There are three main claims in the paper:

1) A new method FIT for flexible and dynamic adjustment of feature focus during inference time.
2) Experiments across several NLP tasks such s sentiment analysis, QA, and NLI.
3) Generalization to unseen features and distribution shifts over feature values.

At a surface level all of the above claims are supported.  The proposed approach allows for adjustment of features during inference time and is evaluated on three datasets focusing on sentiment analysis (SS),  NLI (SMNLI), and QA (BBQ).  As well as an additional dataset called SHANS  for an additional NLI comparison in the appendix (L).

However, I have concerns about the setup and evaluations and whether they support the claims entirely.
(1) I am unconvinced that the method is flexible or dynamic as it requires explicit knowledge of the task and spurious features.  The user need to already know apriori whether something is spurious or not and be able to express it in “ignore X” format. This would require already knowing the answer, to then know whether it should ignore anything outside the context or not as in the given example.
(2) The evaluated tasks are easy tasks for the size of model and do not accurately reflect the current evaluation landscape for such sized models.
(3) Spurious features that are added may not be highly relevant and challenging to the task.  How often is pineapple in the SS dataset?
(4) Evaluations are limited to the same size of model (7B).  It would be good to see models on either end for trends.  Do smaller models have more trouble following these instructions, for example?

**Essential References Not Discussed:**

I believe the authors have done due diligence to cite related work in the paper.  However, they do not run comparison to many of these works, some of which seem to be highly relevant  - those in lines 161+.  They mention being white box methods, however, I think it’s important still to make comparison in order to know if there’s any decrease in performance compared to such methods.  Another fair comparison from related work are the RLHF strategies - e.g. positive and negative sentences with focus.  Given the easy data generation process it also seems a relatively straightforward approach that should be discussed.  The paper also does not mention too much CoT style approaches which could similarly be used to potentially mitigate spurious features as discussed.

**Experimental Designs Or Analyses:**

Yes.  Some concerns that arise are
1) How are the spurious features like Pineapple and Bayesian chosen?
2) Whether the tasks are challenging  enough for the model size.  See questions above for suggested other tasks.
3) There is no demonstration that forgetting does not occur.  It would be good to see that post finetuning performance does not decrease performance on other tasks.  This can be verified on other tasks as is standard in studying the alignment tax.

**Methods And Evaluation Criteria:**

For evaluation criteria, there are two limitations to the proposed criteria: (1) limited datasets and setup, and (2) metrics.

1) The proposed evaluation datasets are commonly used to evaluate NLP models, however these are old evaluations (pre-LLM era) and many newer benchmarks are used to evaluate LLMs.  Given that the paper mostly investigates larger language models and focuses on instruction tuning of LLMs, I would expect evaluations which are closer to the more conventional QA evaluations such as ARC, MMLU, HellaSwag, etc. these tasks pose a harder challenge for LLMs of this scale which may induce different performance for the given approach.
2) The authors propose to use focus accuracy to measure performance of the model.  While I understand why it’s important to evaluate focus accuracy, conventional evaluation of spurious correlations (e.g. in the vision literature such as evaluations over Waterbirds classification, CelebA spurious features, etc.) evaluate both group performances as well as overall accuracy.  I would like to see overall accuracy as an added metric for comparisons.

**Other Comments Or Suggestions:**

N/A

**Other Strengths And Weaknesses:**

One particular strength I want to highlight is the use of example figures (Figures 1, 2).  These help make the paper clear in terms of the problem and examples.

**Questions For Authors:**

How are the spurious features chosen to add to the dataset (for example bayesian, pineapple)? What about other features that might be more closely tied to the task?

The exact numbers from the Figures 3-5 are hard to read. Is it possible to include tables with the full results in the Appendix?

**Relation To Broader Scientific Literature:**

Instruction tuning is one of the premier methods used to specialize LLMs, and LLMs are currently a main focus of the machine learning community.  This paper targets bias and fairness of IT for LLMs, which is an important topic.  The paper has highlighted many works in the space of instruction tuning, aligning LLMs, latent steering, though does not mention bias mitigation and reliance on spurious features in the related work section, though this seems a major focus based on the intro (lines 31+).

**Theoretical Claims:**

I have looked at the proofs in the Appendix I.  The propositions make some strong assumptions about the problem, especially a balanced label distribution, and independence conditions.  I understand this may only be needed for the theory, and the theory is not a focus of the paper as it is not stated in the main text, however authors should add some explanation or justification for if this is needed.

There may also be some minor typos in this section such as Theorem I.1 vs. Proposition I.1 (line 1367).

---

> ### Author Rebuttal · Authors · 2025-03-31
>
> ## P1: SS Dataset, Keyword Feature Choices, and Theoretical Assumptions
> The SS dataset provides a controlled setting to verify FIT’s effectiveness by comparing focus accuracies against theoretical predictions without confounding artifacts present in more complex datasets (e.g., BBQ, SMNLI). These simple-to-complex checks with new methods aligns with common practice in ML literature (e.g., [SNGP](https://arxiv.org/pdf/2205.00403), or see [this survey on causal ML](https://arxiv.org/pdf/2206.15475)). Our theoretical assumptions, such as balanced labels, are practical and commonly used, and although flexible, relaxing them would unnecessarily complicate analysis given our goal to eventually test FIT on realistic scenarios (BBQ, SMNLI).
>
> In adding “Bayesian” and “pineapple” to SST to create SS, we intentionally chose words that are arbitrary and semantically neutral to avoid sentiment alteration or label mismatches during synthesis. [Prior work](https://aclanthology.org/2024.findings-eacl.68.pdf) has shown that even task-irrelevant words (e.g., “Performances”) can act as spurious features in sentiment datasets, motivating our design. FIT’s effectiveness is not dependent on specific keywords; any neutral words would suffice.
>
> ## P2: Dataset Choices and Model Sizes
>
> BBQ was originally tested with UnifiedQA’s 11B model, comparable to or larger than our evaluated models. Although extending FIT to more challenging tasks is a valuable future direction, our current results show even large models struggle (e.g., SMNLI causal accuracies remain below 90% after fine-tuning), highlighting that these tasks are still non-trivial even for current LMs. Moreover, this controlled setting with decent base accuracies achievable allows clear evaluation of FIT’s effectiveness in dynamically switching focus without confounding factors from extremely low accuracy. To further address the difficulty comment, we demonstrate FIT’s effectiveness in a more complex NLG setting, see response P1 to reviewer ivne.
>
> ## P3: On Demonstrating that FIT Does Not Lead to Forgetting
>
> We refer the reviewer to Section 5 (starting at line [391]), where our ablation study on the Alpaca-GPT dataset demonstrates that post-fine tuning with FIT shows no over-specialisation or forgetting.
>
> ## P4: On Model Size Trends for FIT
>
> We agree with the reviewer that investigating model size trends is important. In our work, we have already examined this upward trend, demonstrating strong and consistent performance from 7B to 13B models (note that the Vicuna-13B-v1.5 model used in our experiments is indeed 13B, not 7B). To further illustrate FIT’s adaptability, we have run an additional experiment to evaluate its performance on the Qwen-2.5-3B-Instruct model. For instance, for focus on causal (focus(C)) prompts, we observed a trained accuracy of approximately 96.3% and an unseen accuracy of around 95.7%. For focus on spurious (focus(S)) prompts, the trained accuracy was about 73.9% with an unseen accuracy near 66.0%. These results demonstrate that FIT’s robustness to model size. Full results can be found at [this anonymised document](https://osf.io/5kncm/?view_only=ca4c684c7ee642d7ad7cdcc84c87ea17).
>
> ## P5:  Dynamic Adaptability  and Prior Knowledge
>
> Our experiments on BBQ and SMNLI (Sections 4.2 and 4.3) demonstrate clearly FIT’s dynamic adaptability, enabling models to generalise under distribution shifts and to unseen or shifted features using only provided focus instructions, without pre-identifying spurious features or their spurious labels during inference. While FIT requires initial identification of spurious features for training only, this aligns well with common industry practices for transparency and reliability, can be used with automated methods of spurious feature identification, and doesn’t limit FIT practically  (see our existing detailed discussion in Appendix B).
>
> ## P6: On Comparing to White-Box Methods  and CoT Baselines
>
>  White-box methods discussed in related work aren't directly comparable to FIT, as they require model access (white-box) during both training and testing. Moreover, latent steering methods (LSMs) typically require training per feature, whereas FIT teaches a generalizable capability to adapt to unseen features (demonstrated by our BBQ generalization results). Furthermore, note that recent work ([Wu et al., 2025](https://arxiv.org/abs/2501.17148)) shows LSMs significantly underperform compared to SFT, the stronger baseline that we use throughout our paper.
>
> CoT baseline: Using the base Llama-3.1-8B-Instruct model on BBQ, zero-shot CoT focus accuracy for each prompt type was near random (e.g., ~35% for causal, ~33% for spurious), indicating FIT's superiority.
>
>
> ## Conclusion and Final Comments
>
> We thank the reviewer for their constructive comments, which have helped strengthen our paper. We hope our responses adequately address their concerns and remain available for any further clarifications.

---

### Official Review · Reviewer_ivne · 2025-03-15

**Overall Recommendation:** 3

**Summary:**

This work presents a training method to improve model steerability w.r.t specific features. The main idea is to add instructions during training about what to focus on and what to ignore. The trained model is evaluated on a modified SST dataset, a modified MNLI dataset, and BBQ, and it demonstrates significantly better steerability in terms of following the instruction to avoid spurious features (even unseen ones in the case of BBQ).

**Claims And Evidence:**

The experiments successfully demonstrate the effectiveness of this method. The high accuracies indicate that different instructions can steer the model to different predictions. Additionally, it is really great to see that on the BBQ dataset, even for untrained features, the steerability seems to be improved quite significantly. That said, I still have slight concerns about how generalizable this approach would be (see below).

**Essential References Not Discussed:**

No missing essential references.

**Experimental Designs Or Analyses:**

No major issues.

**Methods And Evaluation Criteria:**

My main concern about this work is that it works in a very clean setting where the testing set and the training set are in similar distributions, and all these features are also defined in a relatively clean setting. Ideally, the goal of steerability is to be able to steer the model in out-of-distribution data where retraining is hard or expensive to do. If the authors can show how the model's general steerability improves on significantly different tasks or even tasks beyond classification, this work will be substantially more impactful.

**Other Comments Or Suggestions:**

I'd like to encourage the authors to bring some of the content on how the training set is constructed from the Appendix to the main paper. I find that part to be quite interesting, and I also feel that some early sections can be compressed.

**Other Strengths And Weaknesses:**

This is in general a well-written paper with clear takeaways. Despite my concern on the further generalizability of this method, I think it already has value in improving steerability in relatively clean and in-distribution settings.

**Questions For Authors:**

n/a

**Relation To Broader Scientific Literature:**

This work proposes a method to improve steerability w.r.t specific features. Debiasing the model through contrastive examples and balancing the training set are well-known ideas. This work is a natural step from those works and leverages instruction following the ability of LLMs and showing good performance in the evaluation.

**Theoretical Claims:**

No theoretical claims.

---

> ### Author Rebuttal · Authors · 2025-03-31
>
> We thank the reviewer for their comments regarding our introduction, specifically noting the “significantly better steerability” and overall “effectiveness” of our method. We hope that this rebuttal response addresses your additional comments regarding our paper.
>
> ## P1: Showing Further Generalisability of Our Method
> We agree that extending beyond classification and MC-QA tasks is highly beneficial for demonstrating that our method can generalise to more complex settings, thereby enhancing the overall impact of our work. To build on our previous results, we have extended FIT to operate in an NLG setting, an environment that poses additional challenges compared to the classification scenarios considered thus far.
>
> **Extension to NLG Setting:**  We adapt our BBQ experiments to formulate a new NLG experiment. In this adaptation, we remove the predefined answer options from each prompt, leaving only a context and a question. Consequently, the model must generate open-ended text responses rather than select from a limited set of choices (e.g., a, b, or c). This change forces the model to independently identify and determine the answer within the context based on the question, making the task harder than its MC-QA counterpart.
>
> **Evaluation Methodology:** : Assessing correctness in NLG tasks introduces challenges beyond those in classification tasks, due to the need to account for semantically equivalent expressions of the correct response meaning. To evaluate model correctness in a computationally efficient manner, we use the Llama-3.1-8b-Instruct model as a judge [1], which determines if the model’s response is semantically equivalent to the reference answer. We manually verified that this approach provides a good measure of semantic equivalence, aligning well with human judgments.
>
> **Focus Accuracy Results:** : We present the focus accuracy results for the same dataset setup as in the original BBQ experiments. Our results include evaluations on both in-distribution (seen) features and out-of-distribution (unseen) features. The trends observed for our Llama model are consistent with those found for the Mistral and Vicuna models.
>
>
> ### LLaMA Focus Accuracy for Seen / Unseen Features
>
> |  | $\emptyset$                    | focus(C)              | focus(C) $\wedge$ ignore(S)  | ignore(S)       | focus(S)             | focus(S) $\wedge$ ignore(C)    |
> |-----------|---------------------------|---------------------------|----------------------------|----------------------------|----------------------------|-----------------------------|
> | Few-shot  | 76.07% / 81.33%           | 73.78% / 79.00%           | 77.67% / 78.67%            | 78.20% / 82.67%            | 44.44% / 39.67%            | 41.69% / 34.00%             |
> | SFT       | **99.31%** / **98.33%**   | 98.93% / **97.67%**       | 98.32% / 97.00%            | 98.32% / **96.67%**        | 23.86% / 23.67%            | 23.55% / 23.33%             |
> | FIT       | 99.24% / 96.67%           | **99.54% / 97.67%**       | **99.62% / 97.33%**        | **99.54% / 96.33%**        | **97.10% / 77.67%**        | **97.41% / 79.00%**         |
>
> *Summary of P1:*  These results indicate that FIT successfully generalises to the more complex NLG setting, exhibiting significant steerability for both in-distribution and out-of-distribution test data. The improvements over the zero-shot, few-shot, and SFT baselines underscore the robustness of our method. Furthermore, we will include the full experimental results in the final version of the paper to further support our generalisability claims. In addition, we plan to modify and extend the future work section of the paper to reflect these updates and to propose new extensions along the lines suggested by the reviewer.
>
> ## P2: On Moving Some of the Content of the Training Set Construction to the Main Paper from the Appendix.
>
> We also recognise the value of moving some of the detailed information regarding the dataset construction to the main section of the paper. Where possible, we plan to integrate additional details into the main paper. The final extent of this inclusion will depend on the available space in the final version; however, we will at least provide further context on the training set construction to enhance clarity.
>
> ## Conclusion and Final Comments
> In summary, our extended NLG experiment demonstrates that FIT can generalise to more complex tasks and perform effectively on out-of-distribution data, thereby following the reviewer’s suggestion, enhancing the potential impact of the paper.
>
> We thank the reviewer for their constructive feedback, which has been useful in further refining our work. We remain available to respond to any further comments or questions.
>
> # References
> [1] Gu, Jiawei, et al. "A survey on llm-as-a-judge." arXiv preprint arXiv:2411.15594 (2024).

---

### Decision · Program_Chairs · 2025-05-01

**Decision:**

Accept (poster)

**Comment:**

The proposed method, Focus Instruction Tuning (FIT), is a variation of supervised fine-tuning (SFT) in which the fine tuning process is conditioned, via a specific templated prompt, so that the model focuses on specific desired aspects while ignoring others. Authors show a few experimental settings in which the model is steered to be robust with respect to spurious features, or so that social biases are mitigated by ignoring features such as age or gender (in tasks where this should be ignored). Results show improvements on specific tasks compared to SFT and zero-shot.

Reviewers have brought up some shared concerned that the authors have addressed in the rebuttal. For example the fact that the experimental evaluation was limited to multiple choices answers. In response during the rebuttal authors have presented results on open ended generation on a modified BBQ task. The correctness of the answers was assessed by LLM as a Judge (Llama 3.1-8b-instruct). Results show that FIT performs comparable to SFT on in-distribution data but outperforms SFT in out-of-distribution data.

Another common initial criticism was related to the fact that the user needs to know in advance what are the spurious features they want the model to ignore. While this is true, the authors pointed out that this is a fairly typical assumption.

There were also some concerns about the fact that the experiment that shows the ability of the model to ignore spurious correlation is a non realistic scenario. The authors defended their choices stating that spurious correlations has been shown to be “effective” even if induced by task-irrelevant words.

The reviewers also pointed out that while fine-tuning is fundamentally different than steering, the setting in which the model is tested would be compatible with some recent steering algorithms so a comparison with some of these algorithms would strengthen the paper. The authors justified the lack of such comparison by stating that activation steering methods rely on having access to the activations of the model while FIT does not. Independently, I believe this comparison would make the paper stronger. However, and more importantly, I do not understand how this method is not a “white box” method, yet requires to compute gradients and change the weights of the model. This seems a contradiction that should be addressed in the camera ready.

There is one remaining major concern that in my opinion has not been fully addressed: the lack of evidence that the overall model utility has not been degraded.

Since this method requires training which eventually lead to changes in the weights it is not something the user can leverage and then discard at the next request. If the model is expected to remain generic, then it is important to assess the potential degradation (or lack of) that FIT might induce to the overall model utility. The community typically measure this via for example change in MMLU and Perplexity (just to mention a few options). Authors have shown a small experiment in which they compare the responses of Alpaca-GPT before and after FIT and showed that the the models remain somewhat aligned (about 3.5 score on a scale from 1 to 5 where 5 means perfect alignment and 1 means perfect misalignment). While this might be a promising clue it remains difficult to interpret what a 3.5 score actually means in term of model utility.

The method is interesting for the community and the reviewers agree on acceptance. Therefore I'd expect the authors to do one of the following for the camera ready:
1. Clearly state that the FIT model is meant to be used in specialized contexts (valid option) but might loose general abilities.
OR
2. Provide experiments about generic model abilities before and after FIT (e.g., MMLU).